# On the estimation of stratospheric age of air from correlations of multiple trace gases

Florian Voet[1], Felix Ploeger[1,2], Johannes Laube[2], Peter Preusse[2], Paul Konopka[2], Jens-Uwe Grooß[2], Jörn Ungermann[2], Björn-Martin Sinnhuber[3], Michael Höpfner[3], Bernd Funke[4], Gerald Wetzel[3], Sören Johansson[3], Gabriele Stiller[3], Eric Ray[5,6], and Michaela I. Hegglin[1,2,7]

[1]Institute for Atmospheric and Environmental Research, University of Wuppertal, Wuppertal, Germany
[2]Institute of Climate and Energy Systems: Stratosphere (ICE-4), Forschungszentrum Jülich, Jülich, Germany
[3]Institute of Meteorology and Climate Research - Atmospheric Trace Gases and Remote Sensing (IMK-ASF), Karlsruhe Institute of Technology, Karlsruhe, Germany
[4]Instituto de Astrofísica de Andalucía, CSIC, Spain
[5]NOAA Chemical Sciences Laboratory, Boulder, CO, USA
[6]Cooperative Institute for Research in Environmental Science, CU Boulder, CO, USA
[7]University of Reading, Department of Meteorology, Reading, UK

**Abstract.** The stratospheric circulation is an important element in the climate system, but observational constraints are prone to significant uncertainties due to the low circulation velocities and uncertainties in available trace gas measurements. Here, we propose a method to calculate mean age of air as a measure of the circulation from observations of multiple trace gas species which are reliably measurable by satellite instruments, like trichlorofluoromethane (CFC-11), dichlorodifluoromethane (CFC-12), chlorodifluoromethane (HCFC-22), methane ($CH_4$), nitrous oxide ($N_2O$) and sulfur hexafluoride ($SF_6$), and show that this method works well in most of the lower stratosphere up to a height of about $25\,km$. The method is based on the compact correlations of these gases with mean age. Methodological uncertainties include effects of atmospheric variability, non-compactness of the correlation, and measurement related effects inherent for satellite instruments. The multi-species age calculation method is evaluated in a model environment and compared against the actual model age from an idealized clock tracer. We show that combination of the six chosen species reduces the resulting uncertainty of derived mean age to below 0.3 years throughout most regions in the lower stratosphere. Even small-scale, seasonal features in the global age distribution can be reliably diagnosed. The new correlation method is further applied to trace gas measurements with the balloon borne Gimballed Limb Observer for Radiance Imaging of the Atmosphere (GLORIA-B) instrument. The corresponding deduced mean age profiles agree reliably with $SF_6$-based mean age below about $22\,km$ and show significantly lower uncertainty ranges. Comparison between observation-based and model simulated mean age indicates a slow-biased circulation in the ERA5 re-analysis. Overall, the proposed mean age calculation method shows promise to substantially reduce the uncertainty in mean age estimates from satellite trace gas observations.

# 1 Introduction

The stratospheric meridional overturning circulation, termed Brewer-Dobson circulation (BDC, see e.g. Holton et al., 1995; Butchart, 2014), is an important element in the climate system as it controls the distributions and variability of long-lived trace gas species in the stratosphere and troposphere. Changes in such radiatively-active species, in turn, may substantially affect the global radiation budget and atmospheric chemistry. Hence, profound knowledge on BDC changes and the dynamical mechanisms involved is crucial for understanding past and future climate variations.

As BDC velocities are approximately three orders of magnitude slower than typical horizontal winds and cannot be directly measured, they must be inferred from the stratospheric distributions of long-lived trace gas species (Butchart, 2014), such as $SF_6$, $CH_4$, $N_2O$, or different chlorofluorocarbons (CFC's). But the relations between changes in these chemical species and changes in the BDC are complex and not fully understood. In particular, changes in observed trace gas mixing ratios do not indicate a long-term acceleration of the BDC, as predicted by global climate models in response to increasing greenhouse gas concentrations (e.g. Waugh, 2009; Engel et al., 2009). Furthermore, stratospheric reanalysis data sets, which are designed to provide a best guess of the atmospheric state, show very different changes in the BDC over past decades (e.g. Chabrillat et al., 2018; Laube et al., 2020; Ploeger et al., 2019, 2021b). Hence, there is a need for improvement in the representation of the BDC in models and reanalyses by use of better observational constraints, and for enhancing process understanding on the effects of the BDC on the global distributions of long-lived trace gases.

Mean age of air (AoA) is a commonly used diagnostic to determine the strength of the BDC from trace gas mixing ratios (Waugh and Hall, 2002). As first described by Hall and Plumb (1994), mean age can be deduced from mixing ratios of a so-called "clock-tracer", a trace gas species which is linearly increasing in the troposphere and is free of sources and sinks in the stratosphere above. In models, such idealized tracers can be easily defined and AoA can be calculated exactly. In the real atmosphere, stratospheric AoA can be derived from observations of trace gas species that approximate ideal clock tracers. Two trace gases that represent suitable approximations of ideal clock tracers are $SF_6$ and, to a lesser extent, carbon dioxide $CO_2$. Since air enters the stratosphere mainly through the tropical tropopause region, the mixing ratio time series of an ideal clock tracer at some point in the stratosphere can be approximated by shifting its tropical tropospheric time series by a specific time. This lag time can be interpreted as the mean age of air at the point of observation in the stratosphere. The lag time can be determined by identifying that point in time at which the tropospheric record matched the observed mixing ratio (Hall and Plumb, 1994). This leads to a strong negative correlation between the mixing ratios of approximate clock tracer and AoA in the stratosphere, because higher mixing ratios can be found later and lower ones earlier in the tropospheric record. The lag time method is mentioned here, because it illustrates the concept of AoA and explains the occurrences of tight correlations between approximate clock tracers and AoA, which build the foundation of this study. However, as $SF_6$ and $CO_2$ deviate from ideal clock tracers in certain specific ways (see below), more sophisticated methods that take at least some of these deviations into account are usually employed instead of a simple lag time. $CO_2$ possesses a strong seasonal cycle in tropospheric mixing ratios superimposing the long-term increase and an additional stratospheric source related to methane oxidation. $SF_6$ mixing ratios

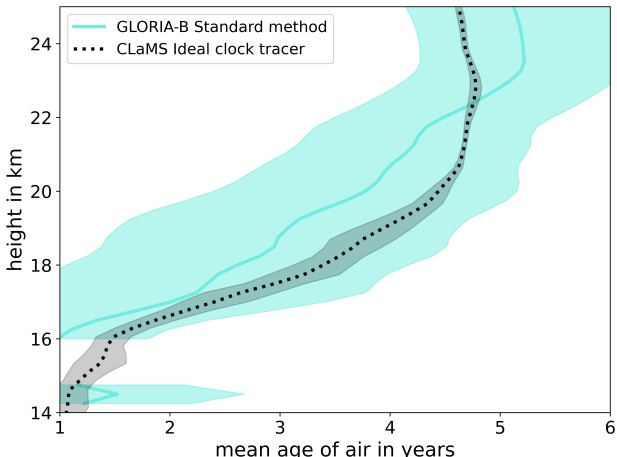

**Figure 1.** Mean AoA/height-profile along measurement locations during the balloon launches of GLORIA-B from Kiruna, Sweden in late August 2021. Blue line: Mean AoA/height-profile calculated with the standard method (Garny et al., 2024b) and sink-correction (Garny et al., 2024a) from $SF_6$ observations of GLORIA-B. Blue shading: Uncertainty range of blue line. Dashed black line: Actual AoA of the CLaMS model from the clock-tracer calculated from the mean values of the ideal model clock tracer along the points of observation by GLORIA-B . black shadings: One standard deviations around the respective mean values that constitute dashed black lines.

in the stratosphere are influenced by downward transport of $SF_6$-depleted air from the mesosphere. The tropospheric evolution of both $SF_6$ and $CO_2$ deviates from perfect ideal clock-tracer linearity.

In addition to these deviations from the behavior of an exact clock-tracer, the retrieval of $SF_6$ and $CO_2$ mixing ratios from observations of remote sensing instruments on satellites or balloons is challenging and comes with significant uncertainties for single profiles (Stiller et al., 2008, 2012; Haenel et al., 2015; Saunders et al., 2024). AoA derived with standard methods from satellite observations of $SF_6$ and $CO_2$ is therefore also subject to high uncertainty, limiting the accuracy of the standard methods to calculate mean age of air for satellite measurements (Waugh and Hall, 2002). In the case of satellite measurements of $SF_6$, for instance, AoA calculations can be realized for zonally averaged data because the density of measurements along the orbit path is usually dense enough to provide enough observations in a latitude band for reducing the measurement uncertainty sufficiently (Stiller et al., 2012).

Figure 1 illustrates this problem using the mean $SF_6$ profile from the remote sensing measurements during a recent balloon flight (see methods and Sect. 3.4). The $SF_6$ measurement uncertainty translates into an age uncertainty of several years. This age uncertainty is so large that it spans the range between the measured and simulated age profiles, although these mean profiles differ by around one year. Hence, the mean age calculated from $SF_6$ remote sensing measurements is insufficient for constraining the stratospheric circulation on a small scale. Therefore, including other trace gas species with lower measurement uncertainty in the mean age calculation appears particularly appealing to decrease the inherent uncertainty (Ray et al., 2014; Engel et al., 2017). Especially for deducing mean age along single measurement profiles, without including additional averaging, such improvements of the methodology to calculate mean age are needed.

Leedham Elvidge et al. (2018) have shown that the trace gases $CF_4$, $C_2F_6$, $C_3F_8$, $CHF_3$, HFC-125a and HFC-227ea also constitute approximations of ideal clock tracers and can be used to derive stratospheric AoA with standard methods. However, the aforementioned six trace gases are similarly difficult to retrieve from remote sensing measurements like $SF_6$ and $CO_2$ and we here focus on other species which can be reliably measured with typical satellite instruments. As a priori values of $CO_2$ are commonly used for the calibration of remote sensing instruments and it is not commonly retrieved at all, $CO_2$ is also not included in this study. In the case of in-situ measurements, Ray et al. (2024) recently proposed a different new method to increase the accuracy of AoA by deriving it from $SF_6$ and $CO_2$ simultaneously, although requiring additional assumptions (e.g. shape of age spectrum).

Even trace gases that approximate ideal clock tracers to a lesser extent than the previously mentioned can show at least partially tight correlations with AoA in the stratosphere, making them potentially useful for determination of AoA from remote sensing measurements, if they can be retrieved with sufficiently low uncertainty (Schoeberl et al., 2005). Even though standard methods to calculate AoA can not be applied for such trace gases, they can still be used for the calculation of AoA, if their correlations with AoA are known (e.g. Hegglin et al., 2014).

This study presents a new method to derive AoA from atmospheric measurements of multiple long-lived trace gases. The method is based on the correlations of the trace gas species $CCl_3F$ (CFC-11), $CCl_2F_2$ (CFC-12), $CHClF_2$ (HCFC-22), $CH_4$, $N_2O$, and $SF_6$ with AoA. The application and evaluation of the method are carried out using simulations with the chemistry transport model CLaMS (Chemical Lagrangian Model of the Stratosphere). Notably, a recent study by Dube et al. (2024) also utilizes the tight correlations between long-lived trace gases and AoA in the CLaMS model. However, their focus is on deriving decadal trends in AoA from these correlations rather than AoA itself. Additionally, Linz et al. (2017) used $N_2O$ in addition to $SF_6$ to calculate AoA. They derived AoA from $N_2O$ using an empirical relationship between AoA and $N_2O$ that was calculated from balloon and aircraft measurements. Their study leveraged AoA to calculate the strength of the total overturning circulation through different isentropes, offering insights into large-scale atmospheric dynamics.

In this study, however, a weighted mean AoA is calculated from modeled satellite measurements of the aforementioned six trace gases. The difference between this weighted mean AoA and the actual AoA from the clock-tracer in the model is found to remain below half a year in the lower stratosphere. In addition, the proposed method is also applied to measurements with the GLORIA-B instrument, yielding more plausible results and drastically reduced uncertainty compared to conventional methods. The present study can be seen as a proof of concept demonstrating how accurately AoA could be calculated from satellite measurements of multiple trace gases and their correlations with AoA. The ESA Earth Explorer 11 candidate Changing-Atmosphere Infra-Red Tomography Explorer (CAIRT) was the primary motivation for the proposed method, but the method is generally applicable to a wider range of data.

The setup of the CLaMS simulation and the proposed method to derive AoA from the correlations of the six trace gases with AoA are described in detail in Sects. 2.1 and 2.2, respectively. Sect. 2.3 describes the GLORIA-B instrument, measurements conducted with GLORIA-B used in this study, and the way AoA was derived for these measurements and compared to CLaMS results. The results of this study are presented in Sect. 3. Here, the new age calculation method is first applied to model data

for a self-consistent "proof-of-concept" and thereafter applied to balloon-borne remote sensing measurements. The results are further discussed in more detail in Sect. 4, and conclusions drawn from this discussion are presented in Sect. 5.

## 2 Methods

### 2.1 Setup of the Chemical Lagrangian Model of the Stratosphere (CLaMS)

CLaMS is an offline Lagrangian chemical transport model, usually driven by wind data from meteorological reanalysis (McKenna et al., 2002b, a). The Lagrangian formulation of transport sets CLaMS apart from most other chemical transport models that operate in an Eulerian framework. As such, CLaMS evaluates the mixing ratios of trace gases species along air parcel trajectories rather than on a fixed spatial grid. The number and distribution of these air parcels change with every time step, forming an adaptive irregular model grid.

The transport in the CLaMS model is based on a hybrid vertical coordinate, $\zeta$, which changes from an orography-following coordinate in the lower troposphere to an isentropic, potential temperature coordinate above the tropopause region and throughout the stratosphere. The model atmosphere is divided by $\zeta$ into 45 different layers. $\zeta$ is equal to the potential temperature $\theta$ above a predefined level, and gradually turns to zero at the surface. Independent of the elevation of the surface, $\zeta$ is defined in a way that it is equal to zero at every surface point (for details see Pommrich et al., 2014). The lowest model layer, representing the lower boundary layer of the model, extends from the surface to approximately 1.5 km (specifically, $0\,\text{K} < \zeta < 70\,\text{K}$), which roughly corresponds to the height of the planetary boundary layer. The uppermost model layer (upper boundary layer) covers the potential temperature range $2350 - 2650\,\text{K}$ (altitude of about 55 km ), hence the model domain extends from the surface to about the stratopause. The emissions of a trace gas over the course of the simulation are specified through its lower boundary condition as mixing ratios. In this study, the lower boundary condition of each trace gas is defined by a set of observation-based time series of zonal mean surface mixing ratios at different latitudes over the simulation period. The values of these time series are interpolated latitude-wise onto the Lagrangian air parcels inside the lower boundary layer at the beginning of each new simulation time step. The air parcels in the upper boundary layer are treated in the same way, with an upper boundary condition on trace gas mixing ratios prescribed in the uppermost model layer. The influence of the upper boundary condition strongly depends on the properties of the given trace gas species in question.

Six suitable trace gas species with long stratospheric life times and sources exclusively in the troposphere were selected for this study. These species are listed in Tab. 1. The tropospheric sources of these trace gases are represented by the lower boundary conditions, since none of them are products of chemical reactions in the atmosphere to any significant amount. The observation data sets used to create the lower boundary condition of each trace gas are referenced in Tab. 1. The atmospheric losses of $N_2O$, $CH_4$, CFC-11 and CFC-12 are taken into account as described by Pommrich et al. (2014). For the use in the present study, the additional trace gas species $SF_6$ and HCFC-22 have been implemented into the CLaMS model similar to the method described by Pommrich et al. (2014). The depletion mechanisms for HCFC-22 implemented in CLaMS is represented by the reactions

**Table 1.** Trace gases implemented in CLaMS used to derive AoA. Lower boundary values of tracers defined with ground-based in-situ measurements

| trace gas | datasets used to create lower boundary |
|---|---|
| Sulfur Hexafluoride (SF$_6$) | Dutton et al. (2023) |
| Trichlorofluoromethane (CCl$_3$F), short: CFC-11 | Dutton et al. (2022b) |
| Dichlorodifluoromethane (CCl$_2$F$_2$), short: CFC-12 | Dutton et al. (2022c) |
| Chlorodifluoromethane (CHClF$_2$), short: HCFC-22 | Montzka et al. (2015) |
| Methane (CH$_4$) | Lan et al. (2023) |
| Nitrous Oxide (N$_2$O) | Dutton et al. (2022a) |

$$CHClF_2 + O(1D) \rightarrow products \tag{R1}$$

$$CHClF_2 + OH \rightarrow products \tag{R2}$$

$$CHClF_2 + h\nu \rightarrow products \tag{R3}$$

for wavelengths $\lambda = \frac{c}{\nu} \leq 220\,\mathrm{nm}$. The photolysis rates for the last reaction are calculated with the CLaMS photolysis code (Becker et al., 2000; Burkholder et al., 2019). Together, these three loss reactions form a significant sink in the stratosphere.

SF$_6$ is the only trace gas in the list that does not have any significant chemical or photolytic loss reaction throughout the troposphere and stratosphere (the here considered model domain). SF$_6$ is mainly removed from the atmosphere through photolysis and electron attachment in the mesosphere, which is not included in the model. Therefore, SF$_6$ is treated simply as an inert tracer. However, depleted SF$_6$ mixing ratios transported downward from the mesosphere into the stratosphere are accounted for by the specification of a realistic upper boundary condition, which is based on measurements from the Michelson Interferometer for Passive Atmospheric Sounding (MIPAS) instrument that operated on the ESA Environmental Satellite (Envisat) from 2002 till 2012 (Fischer et al., 2008). Specifically, the spectra version V5H with the baseline version 21 and the spectra version V5R with the baseline versions 224 and 225 were used (Kiefer, 2021). From these measurements, monthly zonal mean time series of SF$_6$ mixing ratios over the course of the measurement period were created at different latitudes for the potential temperature level 2500 K. The time series for higher latitudes show strong seasonal cycles caused by the downward transport of SF$_6$–depleted air from the mesosphere and the seasonality of the BDC itself. This set of time series is used as the upper boundary for SF$_6$ when the simulation time is within the time period of the measurements. The upper boundary condition for times outside of the measurement period was created by parameterizing the depicted seasonal cycle of

each latitude with a sinusoidal least square fit of a single frequency and adding it to a shifted tropospheric tropical time series (taken from the lower boundary of $SF_6$). The time shifts correspond to the zonal mean age of air values at the respective latitude at the $2500\,K$ potential temperature level. These mean age of air values were derived consistently from MIPAS measurements of $SF_6$ (Stiller, 2021b). This simple parameterization could be further improved by including other modes of variability semi-annual oscillation (e.g. semi-annual oscillation, Garcia et al., 1997) which could be important for analysis of inter-annual variability, that is, however, not considered in this study. Similar to Pommrich et al. (2014), the climatology of the Halogen Occultation Experiment (HALOE, Grooß and Russell, 2005) was used as the upper boundary for $CH_4$ in this study. More specifically, the mean seasonal cycle from the HALOE climatology was used for every year of the simulation period. The Mainz photochemical 2D model (Grooß, 1996) was used for the upper boundary of $N_2O$, CFC-11 and CFC-12 are decomposed completely well below about $2500\,K$ and don't reach the upper boundary layer to any significant amount. Therefore, their upper boundaries were kept at zero throughout the entire simulation period. HCFC-22 does reach the upper boundary layer, but in relation to its tropospheric value in much smaller quantities than $SF_6$. Mesospheric loss processes for HCFC-22 can therefore be neglected compared to the stratospheric ones (reactions R1 to R3). An open upper boundary condition (null flux) has therefore been defined for HCFC-22.

In addition to the six actual trace gases, a completely passive tracer without any atmospheric sources or sinks and linearly increasing lower boundary values was included into the model as well. It will be called the "clock-tracer" in the following and it will be used to calculate the actual mean age of air in the model using the standard lag time method described by Hall and Plumb (1994). The simulation was driven by meteorological data from the European Centre for Medium-Range Weather Forecasts Reanalysis v5 (ERA5) and covered the period from January 1, 1979, to December 31, 2022. The vertical cross-isentropic model transport in CLaMS was calculated from the ERA5 total diabatic heating rates (see Ploeger et al. (2021b)). For the initial condition of the simulated atmosphere, approximately 1.4 million air parcels were spread out by employing the principle of distributing air parcels based on entropy and static stability, as outlined by Konopka et al. (2012) and Pommrich et al. (2014). The mixing ratios of the six trace gases inside the initial air parcels were set to zero at the first time that is 1th January 1979. Thereafter, a 20 year long spin-up simulation has been carried out with repeating lower and upper boundary conditions and meteorology from the first simulation year 1979. This spin-up guarantees that the initial mixing ratio distributions of CFC-11, CFC-12 and the clock tracer at the first day of the transient simulation on 1 January 1979 represent the actual atmospheric conditions reasonably well. The initial mixing ratio distributions of $SF_6$, $CH_4$, $N_2O$ and HCFC-22 were estimated in a different way. The mixing ratios of these four trace gases in every air parcel on 1 January 1979 were all taken from mean tropical time series that were created from the respective lower boundary conditions. The points in time at which the mixing ratios were taken correspond to estimations of the points in time each air parcel was in contact with the surface for the last time before being transported to its final position on 1 January 1979. The estimations are based on annually averaged AoA values derived from observations of $SF_6$ by MIPAS (Stiller, 2021a).

Simulation output for the year 2011, which is near the end of the operational period of MIPAS, was chosen for the analysis of the modeled satellite measurements. At the end of the operational period of MIPAS, the measurement-based upper boundary values of $SF_6$ had the maximal amount of time to spread below the upper boundary in the model atmosphere. The model output

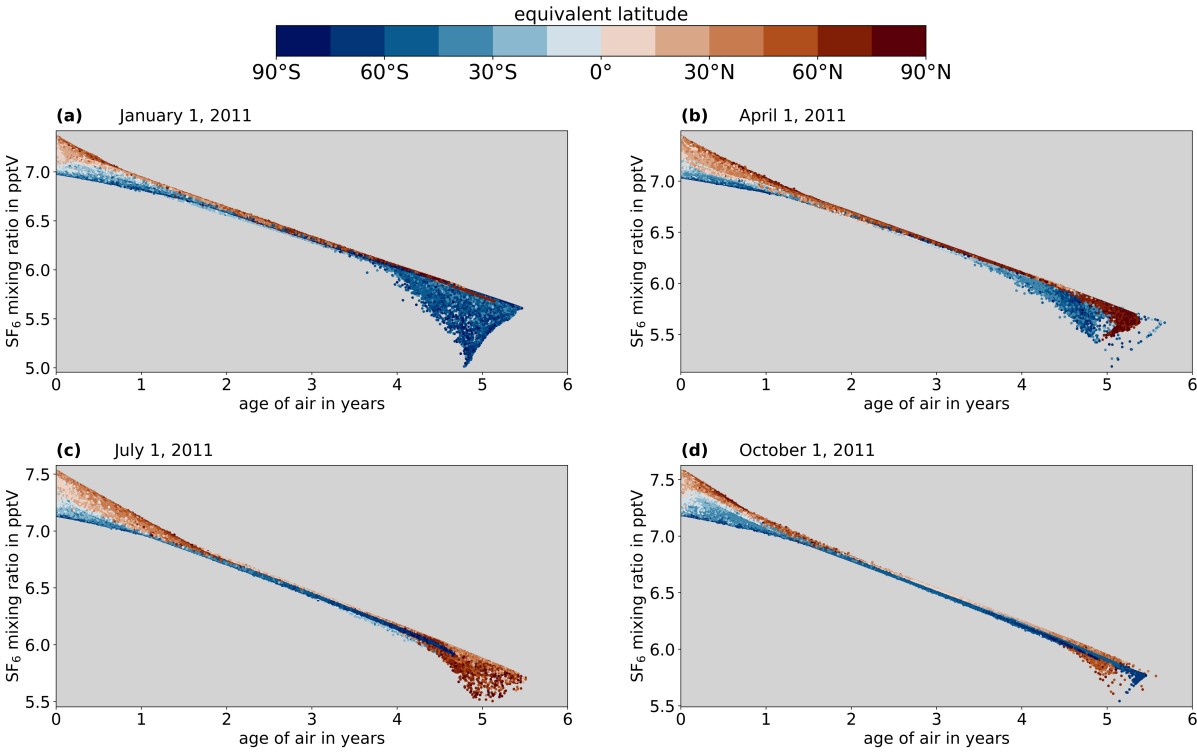

**Figure 2.** Schematic representation of method used to estimate the AoA corresponding to measured mixing ratios and its associated uncertainty (see text for details)

of $SF_6$, which is of particular importance for the aforementioned analysis, is therefore most representative of the real $SF_6$ distribution in the atmosphere for the year 2011. For the application of the proposed method on the GLORIA-B measurements, the CLaMS model output for the days of the two balloon launches was used. The version of CLaMS used in this study is similar to the one described by Konopka et al. (2022) with the CAPE (convective available potential energy) triggering the unresolved convection in the model instead of the moist Brunt–Vaisala frequency. Both parameters as well as the convection parameterization are described in Konopka et al. (2019).

## 2.2 Age of air calculation

The six trace gases were selected on the basis of their long stratospheric lifetimes (see Table 2), for which strong correlations between mixing ratio and AoA can be expected in the stratosphere. These correlations were visualized by taking the model output for individual days and plotting the mixing ratios of each trace gas against the actual AoA values obtained from the model clock-tracer (for a schematic see Fig. 2, for an application to real data see Fig. 3). In order to investigate the influence of seasonality in stratospheric transport, one day from each of the four seasons was selected. Note that the calculation was performed on the original Lagrangian model output (irregular grid) to avoid smoothing due to interpolation. To ensure compact

correlations for all considered species, an upper limit for applicability of the method of $25\,\mathrm{km}$ was established (see Sect. 3). As the dependencies of the mixing ratios with respect to AoA are slightly different for the two hemispheres due to the asymmetry of the BDC (e.g. Konopka et al., 2015), the two hemispheres are considered separately in the following calculations.

Figure 2 illustrates how the model results are used to identify a given tracer mixing ratio with a corresponding AoA value and how the uncertainty range of that AoA value is estimated. The small black dots represent a scatter plot of the modeled mixing ratio against the actual model AoA at all grid points below $25\,\mathrm{km}$ in the same hemisphere and on the same day for any of the six tracers. The entire range of mixing ratios covered by the trace gas at all grid points is divided into 150 bins of the same width, with bin number one being the lowest, and bin number one hundred fifty being the highest in terms of mixing ratio. For visual clarity, only the last three of these mixing ratio bins are shown in the figure. The histogram illustrates the distribution of AoA within a given mixing ratio bin. The blue area in the histogram highlights the one sigma range around the mean AoA value of the distribution (mean value $\pm$ one standard deviation). Similarly, the blue area in each of the three depicted mixing ratio bins corresponds to the one sigma range around the mean value of the respective AoA distribution inside. Such one sigma ranges are calculated for each one of the one hundred fifty mixing ratio bins. Subsequently, a midpoint for each bin with the sample mean AoA as the x- and the middle of the bin range as the y-coordinate was then defined. This constructed set of midpoints constitutes a sort of look-up table that can be used to interpolate an AoA value for any given mixing ratio within the range covered by the bins. Since the binning and averaging of the model data creates some unwanted noise, the series of midpoints is smoothed with a polynomial fit through a Savitzky-Golay-Filter before it is used as a look-up table in the described way. The Savitzky-Golay filter (`scipy.signal.savgol_filter`) was implemented using the SciPy library (Virtanen et al., 2020). The green line in figure 2 represents such a polynomial fit, i.e. the final look-up table. Since the number of points near the edges of the range can sometimes be too sparse to guarantee a sufficient statistic, the first and last couple of bins are ignored during the described procedure, if any of them contained fewer than 50 points.

Depending on the trace gas and the day of the year, the correlation between tracer mixing ratio and AoA can eventually break down as mixing ratios decrease. This means that the gradient of the polynomial fits (lookup tables) can become extremely steep, resulting in essentially no variation in AoA for a varying mixing ratio (see Fig. 4). Lower mixing ratios, where the gradient becomes too steep, no longer carry information about AoA and are therefore no longer useful for its calculation. We consider the correlation of a trace gas with AoA to be at risk of breaking down if the gradient of the respective lookup table exceeds five times the average gradient in the AoA range between one and four years. If this condition is met in five out of nine consecutive mixing ratio bins, the correlation is considered to have broken down, and the series of bins in question, as well as all subsequent bins, are removed from the lookup table. This ensures that short intervals of steeper gradients are still included in the lookup tables.

The uncertainty of AoA obtained from interpolating the lookup table at a measured mixing ratio is based on the spread of the clock tracer AoA around the mixing ratio (model variability) and the uncertainty of the mixing ratio itself that stems from its measurement. For the model variability, the aforementioned standard deviation of the AoA distribution in the respective mixing ratio bin is used. For the measurement uncertainties of the six trace gases, the values listed in Tab. 2 are used. These values represent estimations of the total uncertainties for the measurement of the six trace gases with the proposed CAIRT

instrument. These values are representative for a typical satellite instrument and take into account both the detector noise as well as biases e.g. from in-flight calibrations (ESA, 2023). It is expected that this is a reasonable reflection of the true
uncertainty as long as the AoA inferred from several trace species are combined and the uncertainty would be combined, but likely is an underestimate where only one or two trace-species remain to have meaningful correlations. The conversion of the mixing ratio measurement uncertainty into the uncertainty of the obtained AoA is done through error propagation by calculating the tangent to the polynomial fit at the measured mixing ratio (see Fig. 2). The model variability and the converted measurement uncertainty added together as a Root Mean Square Error (RMSE) yield the total uncertainty of the AoA obtained
for the measured mixing ratio. Instead of doing the described calculation of the total uncertainty each time an AoA value is interpolated from the lookup table, the uncertainty calculation is done in advance for every point of the lookup table and the series of resulting uncertainties is subsequently added to the table. This way, both AoA and its uncertainty can be interpolated from the lookup table for a given mixing ratio.

**Table 2.** Expected measurement error and atmospheric (stratospheric) lifetime for each trace gas

| Trace Gas | Expected measurement error [a] | Total atmospheric lifetime (years) [b] | Stratospheric lifetime (years) [c] |
|---|---|---|---|
| $SF_6$ | 0.2 ppt | 850–1280 | – |
| CFC-11 | 20 ppt | 52 | 55 |
| CFC-12 | 20 ppt | 102 | 103 |
| HCFC-22 | 6 ppt | 11.6 | 120 |
| $CH_4$ | 0.04 ppm | 11.8 | 123 [d] |
| $N_2O$ | 1 ppb (< 20 km) 4 ppb (> 20 km) | 109 | 195 [d] |

[a]  Based on the analysis described in chapter 4.3.5 of European Space Agency report of assessment (ESA, 2023).

[b]  *WMO (2022) Total Lifetime* in Table A-5 in Annex of (WMO, 2022).

[c]  *Stratospheric Lifetime 2022* in Table A-5 in Annex of (WMO, 2022).

[d]  (Brown et al., 2013).

The method described above can be used to derive an $AoA_i$ with a respective uncertainty $\sigma_i$ for each one of the six trace
gases $i$, if all of them are measured simultaneously. In this case, the uncertainty-weighted mean AoA

$$\overline{AoA} = \frac{\sum_i \left( \sigma_i^{-2} AoA_i \right)}{\sum_i \sigma_i^{-2}} \tag{1}$$

and the uncertainty thereof

$$\overline{\sigma} = \sqrt{\frac{1}{\sum_i \sigma_i^{-2}}} \tag{2}$$

can be calculated. The weighted mean uncertainty $\overline{\sigma}$ is significantly lower than each one of the individual uncertainties $\sigma_i$.

## 2.3   GLORIA-B instrument and data analysis

GLORIA-B is a cryogenic limb-imaging Fourier-Transform spectrometer. This instrument operates in the thermal infrared spectral region between a wavelength of about 7 and 13 $\mu$m (wavenumber between about 750 and 1400 $\mathrm{cm}^{-1}$) using a two-dimensional detector array (Friedl-Vallon et al., 2014; Riese et al., 2014). The high spectral sampling of $0.0625\,\mathrm{cm}^{-1}$ allows the separation of individual spectral lines from continuum-like emissions such that many trace gases can be retrieved simultaneously from the measured spectra with high spatial resolution. Two flights with GLORIA-B have been carried out so far in the framework of the European Union Research Infrastructure HEMERA. The first one was performed over northern Scandinavia from Kiruna (67.9° N, 21.1° E) on 21/22 August 2021. Radiance spectra obtained during the float of the gondola (up to 36 km) were analyzed between 17:11 UTC and 05:39 UTC of the following day. The second flight took place on 23/24 August 2022 over Timmins, Ontario, Canada (48.6 °N, 81.4° W). Spectra were obtained at the beginning of the float of the gondola at 20:35 UTC until 08:26 UTC of the next day. Unfortunately, a leakage in the balloon envelope caused the gondola to sink from 36 km (at the beginning of the float) down to 22 km at the end of the observation period. Retrieval calculations of trace species were performed with a least squares fitting algorithm using analytical derivative spectra calculated by the Karlsruhe Optimized and Precise Radiative transfer Algorithm (KOPRA, Stiller et al., 2002; Höpfner et al., 2002). For a more detailed description of the retrieval method, refer to Johansson et al. (2018). A Tikhonov-Phillips regularization constraining with respect to a first derivative a priori profile was adopted. The resulting vertical resolution is better than $2\,\mathrm{km}$ for all molecules used in this study, except $SF_6$ (better than $3\,\mathrm{km}$) and CFC-113 (better than $4\,\mathrm{km}$). Cloud-affected spectra were filtered out by a cloud index according to Spang et al. (2004). The error estimation of the retrieved target gases includes random noise, temperature errors, calibration, pointing and field of view inaccuracies, errors of non-simultaneously fitted interfering species, as well as spectroscopic data errors. All errors refer to the 1-$\sigma$ confidence limit.

The final product retrieved from the radiance spectra observed during a flight consists of a time series of mixing ratio/height profiles for each of the six trace gases. Each profile in a time series contains the mixing ratio of the respective trace gas on 130 altitude levels in 0.25 km steps from the upper troposphere up to flight level. The altitude levels at which the trace gases are retrieved stay the same over the course of the flight. The latitudes and longitudes of the observation points change over time and along the altitude levels due to the movement of the gondola and the line-of-sight of the instrument. During both flights, the path of the gondola covered a relatively short distance only of approximately 300 km. The CLaMS model results have been interpolated onto the measurement locations of the two balloon flights, for both the modeled mixing ratios of the six tracers and clock-tracer mean age. The interpolation on the points and times of GLORIA-B measurements has been realized by trajectory mapping of the measurement locations on the closest synoptic time where model output is available, and subsequent spatial interpolation. Thereafter, the time series for both flights and both CLaMS and GLORIA-B were temporally averaged, creating mean mixing ratio/height profiles for the six trace gases and the clock tracer. The new correlation-based mean age calculation method described in Sect. 2.2 is then applied to the mean profiles of the observed and modeled trace gases to create AoA/height profiles. Note that the required lookup tables were created from CLaMS model output for the days of the balloon

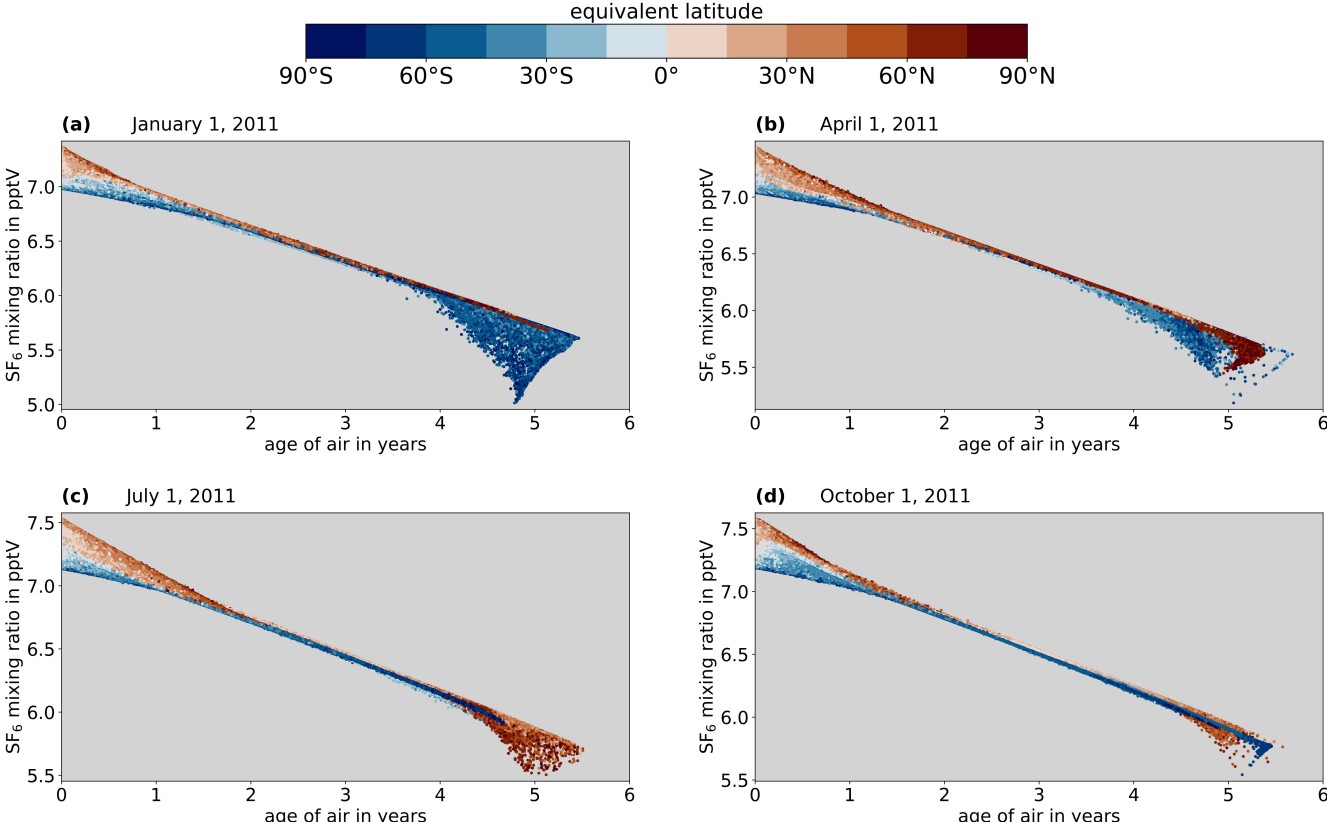

**Figure 3.** Results of CLaMS simulation up to 25 km at four different days in 2011, SF$_6$ volume mixing ratios plotted against mean age of air from clock tracer with color-coded equivalent latitude.

launches. In addition to that, the convolution method described by Garny et al. (2024b) in combination with the correction method described in Garny et al. (2024a) is used to derive AoA/height profiles from just SF$_6$. The actual AoA/height profiles for the model were created from the clock tracer profiles as lag time (Hall and Plumb, 1994).

## 3 Results

### 3.1 Relations between trace gas mixing ratios and age of air

Fig. 3 shows the relation of SF$_6$ to AoA from the results of the CLaMS model for the four selected days in 2011. Note that all panels of the figure show the results for both hemispheres together. For each day shown, the SF$_6$ mixing ratios are plotted against their corresponding AoA values from the clock tracer for all model points below 25 km. Additionally, the equivalent latitudes of the model points (Nash et al., 1996) are color-coded to visualize their dynamical characteristics. The SF$_6$ mixing ratios behave very similarly with respect to AoA at all four times up to ages of roughly 4.5 years. Overall, SF$_6$ mixing ratios

decrease with increasing AoA, which is consistent with the almost linear increase of mixing ratios in the troposphere and the lack of sources and sinks in the stratosphere. There is a significant spread of mixing ratios in young air masses in the same hemisphere for ages below about 1.5 years, i.e. those air masses located in or near the troposphere. Air parcels from the northern hemisphere also have consistently higher mixing ratios than those from the southern hemisphere in this AoA range. This hemispheric difference can be explained by the meridional gradient of $SF_6$ mixing ratios in the troposphere, which is caused by the fact that $SF_6$ emissions occur predominantly in the northern hemisphere and that tropospheric $SF_6$ mixing ratios increase rapidly (Schuck et al., 2024a). From around 1.5 to 4.5 years, $SF_6$ and AoA correlate strongly and their relation appears to be almost perfectly linear. Depending on the day, mixing ratios in either the northern or the southern hemisphere start to spread out for AoA values above about 4.5 years. This collapse of the correlation is predominantly caused by the break-down of the polar vortex at the end of winter in the respective hemisphere, which, in turn, induces downward transport of $SF_6$-depleted air masses from the mesosphere into the mid-latitude stratosphere. Moreover, the spread in mixing ratios is stronger in the southern hemisphere, because the polar vortex is more pronounced in the Antarctic than in the Arctic.

Fig. A1 in the appendix is similar to Fig. 3, but for CFC-12 instead of $SF_6$. Overall, the spread appears to be stronger and the general shape of the correlation is more convex for CFC-12 compared to $SF_6$. The higher compactness of the correlations for $SF_6$ can be explained by the presence of a stratospheric sink for CFC-12 in contrast to $SF_6$ and the resulting much higher atmospheric lifetime of $SF_6$ compared to CFC-12 (see table 2). In air masses in or near the troposphere, the spread is much smaller for CFC-12, because unlike $SF_6$, CFC-12 has no strong meridional gradient in the troposphere around the year 2011. The relatively weak meridional gradient observed for CFC-12 can be attributed to its significantly reduced emissions by 2011, following its global ban in 2010. Similar illustrations of the correlations for the remaining four trace gases can be found in Figs. S1 to S4 in the supplement.

### 3.2 Lookup tables to get AoA from mixing ratio

The considered stratospheric air masses from the CLaMS simulation for the four selected dates were separated into the two hemispheres by use of equivalent latitude. Thereafter, the procedure described in Sect. 2.2 has been applied to the dataset of each trace gas, day and hemisphere to create the look-up table required to identify a given mixing ratio with a corresponding AoA and uncertainty thereof.

Figure 4 shows the results of the aforementioned procedure in the case of CFC-12 for the southern hemisphere. Figure A2 in the appendix shows the corresponding results for the northern hemisphere. The green dots represent the scatter plots of mixing ratio drawn against AoA up to $25\,\text{km}$. The transparent black bars mark the one sigma environments around the mean AoA values of the mixing ratio bins (model variability). Note that in some cases the series of black bars do not extend all the way down from the upper to the lower end of the mixing ratio range (e.g. Fig. 4 (a)). This is because one or both of the conditions described in sect. 2.2 were met (data points become to sparse and/or gradient becomes to steep) and the respective streak of bins at the beginning or end of the mixing ratio range were ignored during the procedure. The conditions described in Sect. 2.2 ensure that only mixing ratio bins with sufficiently large sample sizes in regions of sufficiently tight correlation with AoA are included into the lookup tables.

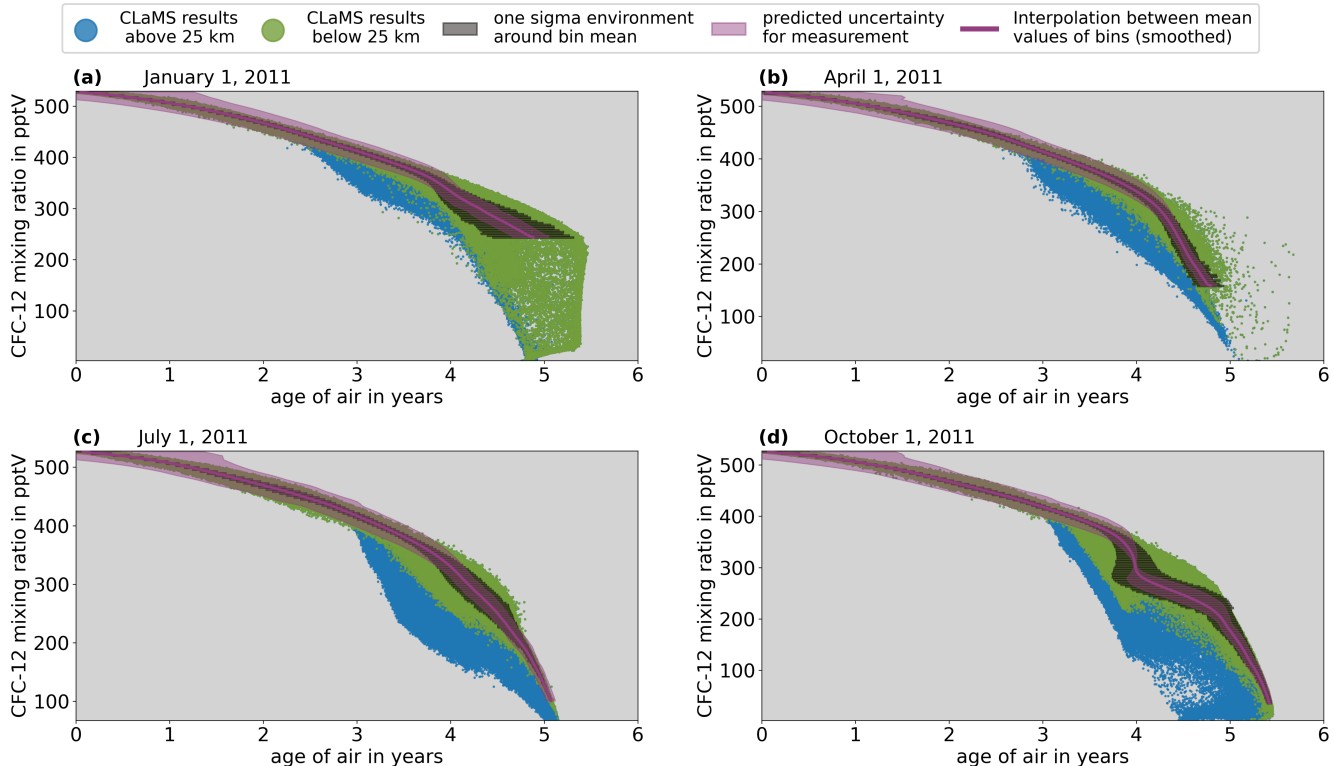

**Figure 4.** Green dots: Results of CLaMS simulation up to 25 km in southern hemisphere (equivalent latitude < 0) at four different days in 2011, CFC-12 volume mixing ratios plotted against clock tracer AoA. Blue dots: Same as green dots, but in altitude region between 25 km and 30 km. Black boxes: One sigma environment around average AoA in volume mixing ratio bins included in lookup table (part of uncertainty of AoA that arises from spread of data points). Magenta line: Series of bin midpoints (mean AoA/mean mixing ratio) smoothed with Savitzky-Golay-Filter (used as lookup table). Magenta shading: Part of uncertainty of AoA that arises from measurement of trace gas (estimated with expected uncertainty in table 2 according to method illustrated in Fig. 2.)

The magenta lines in the two figures show the smoothed series of bin midpoints that constitute the lookup tables for AoA. The magenta shading around each of those lines is the uncertainty range of AoA that results from the expected measurement uncertainty (see Tab. 2). From here on, the uncertainty of AoA that results from the measurement uncertainty of the trace gas, will itself simply be referred to as the measurement uncertainty. The blue dots are the same as the green ones, but in the altitude range between $25\,\mathrm{km}$ and $30\,\mathrm{km}$. The comparison between the green and blue dots illustrates the reason for the general altitude limit of $25\,\mathrm{km}$ for all days and trace gases. A general height limit was chosen only for the sake of simplicity, and $25\,\mathrm{km}$ as the general height limit seems to assure lookup tables of a reasonable length in all cases. In the case of CFC-12 in the southern hemisphere, for instance, the correlation between tracer mixing ratio and AoA probably wouldn't have been compact enough to create a lookup table for October beyond AoA around 4 years, if data between $25\,\mathrm{km}$ and $30\,\mathrm{km}$ had been included (see Fig. 4 (d)). The complete lookup tables (including measurement uncertainty and model variability) for the remaining five trace

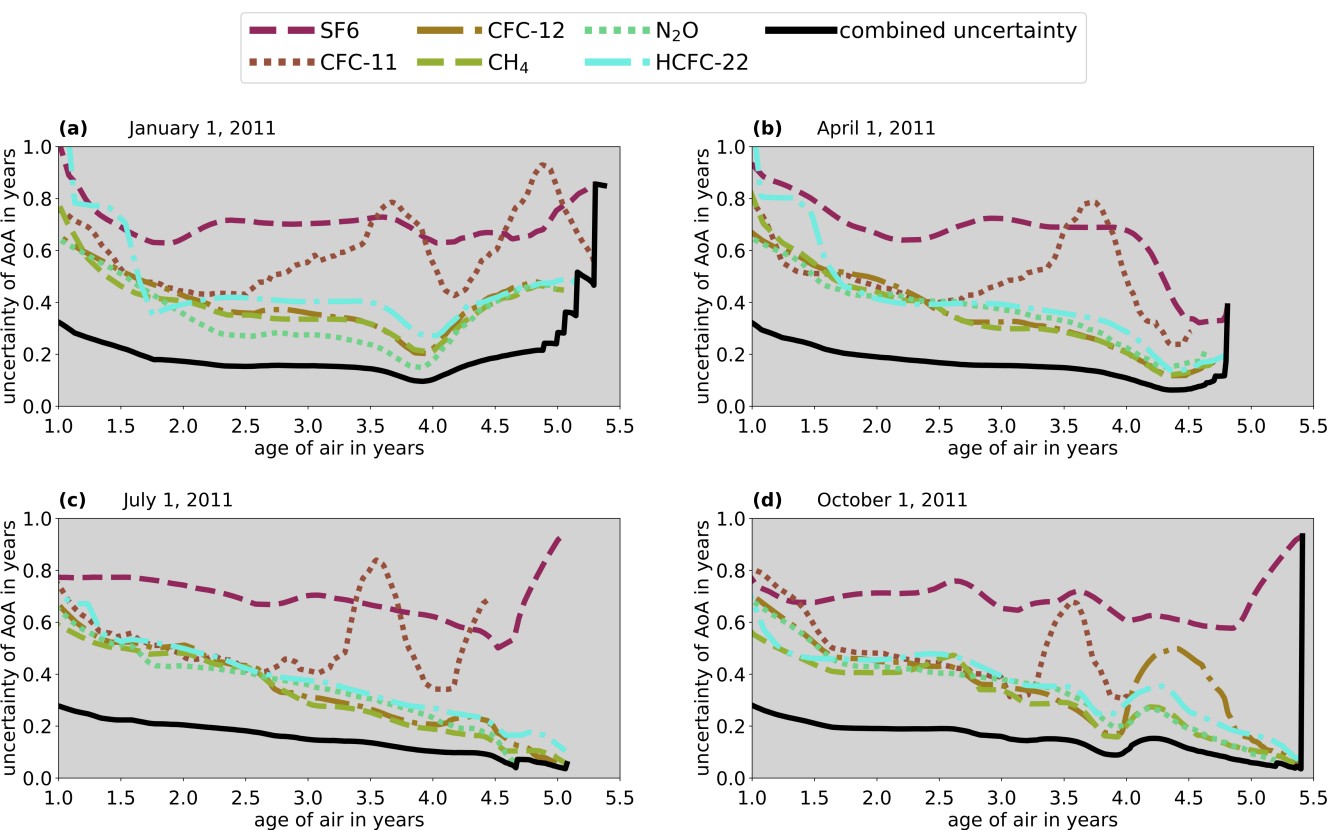

**Figure 5.** Dashed/dotted/dash-dotted colored lines: Individual total uncertainties of the six trace gases in the southern hemisphere at the four considered days in 2011 over the AoA ranges of the respective lookup tables. Black lines: combined total uncertainty calculated with formula 2 from the individual total uncertainties.

gases for the four days and two hemispheres were created accordingly and similar illustrations of them can be found in the supplement (Figs. S5 to S14).

In the case of actual measurements, AoA and its total uncertainty would be interpolated from the lookup tables for each of

345 the six trace gases separately and the weighted mean $\overline{\text{AoA}}$ (see Eq. 1) with the combined uncertainty $\overline{\sigma}$ (see Eq. 2) would subsequently be constructed from the results. Before calculating the actual mean AoA values, an initial estimate of the combined uncertainty for AoA values ranging from one to five and a half years was made. This involved assuming that a specific AoA value within the range was uniformly derived from the lookup tables of all six trace gases, implying that all tracers yielded the same AoA value. This value, treated as the weighted mean AoA, allowed for interpolation of corresponding total uncertainties

from each tracer's lookup table. These uncertainties were then used to construct a combined uncertainty $\overline{\sigma}$.

Figure 5 shows the individual total uncertainties of the six trace gases as well as the combined uncertainty over an AoA range between one and five and a half years for the four days on the southern hemisphere. Fig. A3 in the appendix shows the corresponding results for the northern hemisphere. In every case, the combined uncertainty starts at young ages with a peak

value between 0.2 and 0.4 years and then gradually decreases with increasing AoA up to around at least 4 years. A gradual increase of the uncertainties for AoA higher than about 4 years is the result of the increasing spread in some of the underlying trace gas species data, which in turn occurs due to depleted mixing ratios at higher altitudes. Higher uncertainties at the lower end of the AoA range can be explained by the flatter gradients of the fitted polynomials. The combined uncertainty stays well below 0.3 years over most of the AoA range in all cases. At ages larger than about 4.5 years, sudden steep jumps can be seen in the combined uncertainty. These jumps occur whenever the compact relation between the respective trace gas species and age breaks down (AoA values in the lookup table have reached their limit) and the tracer in question does no longer contribute to the weighted mean and its uncertainty.

To further investigate the atmospheric regions where the presented method provides AoA with a given uncertainty, the combined uncertainties from Figs. 5 and A3 are presented as distributions in the latitude-pressure plane. For this mapping, the actual model AoA from the clock-tracer in the CLaMS simulation was used to project the uncertainties into the latitude-pressure plane. In a first step, zonal mean fields of model mean age were calculated, and these mean age fields are shown in Fig. 6 for the altitude range between 10 and $25\,\mathrm{km}$. These zonal mean values were treated as if they were measurement-derived weighted mean AoA values and were associated with the corresponding combined uncertainty at the respective day and hemisphere depicted in Figs. 5 and A3.

The created zonal mean distributions of the combined uncertainties at the four days are shown in Fig. 7. The white areas in the figure correspond to the regions of zonal mean AoA below one year (compare Fig. 6). Stratospheric air younger than roughly one year is located close to the troposphere and may therefore likely be influenced by small amounts of tropospheric air that entered the stratosphere through the extratropical tropopause (Hauck et al., 2020). This is, for example, evident in the comparatively large spread of $SF_6$ mixing ratios for AoA below roughly one year in Fig. 3. The influence of well-mixed tropospheric air can also lead to flat gradients in the required lookup tables, which in turn lead to a high uncertainty of AoA below one year (see Fig. 4 and A2). Since air masses influenced by the troposphere are not the focus of this study, and the high uncertainties of their AoA values would only distract from the purely stratospheric air masses, AoA below one year is excluded in Fig. 7 and in all following figures.

As expected from the behavior of the combined uncertainties in Figs. 5 and A3, the uncertainties stay well below 0.3 years in all the remaining regions of AoA above one year and tend to decrease with increasing AoA. On average the uncertainties appear highest for austral summer season (1 January) at high southern latitudes, which is likely due to the collapse of the Antarctic polar vortex a few months prior, and the subsequent gradual spread of air masses that were contained in the vortex before its collapse.

### 3.3 AoA from synthetic measurements

Figure 7 represents an estimate how accurately zonally averaged AoA might be derived from trace gas measurements via the proposed correlation method (see Sect. 2.2) in different regions. In the following, we evaluate the method even more thoroughly by creating pseudo-measurements of the six trace gases from the CLaMS model results and using these pseudo-measurements to calculate AoA with the correlation method. These calculated AoA values will be compared to the actual model AoA from

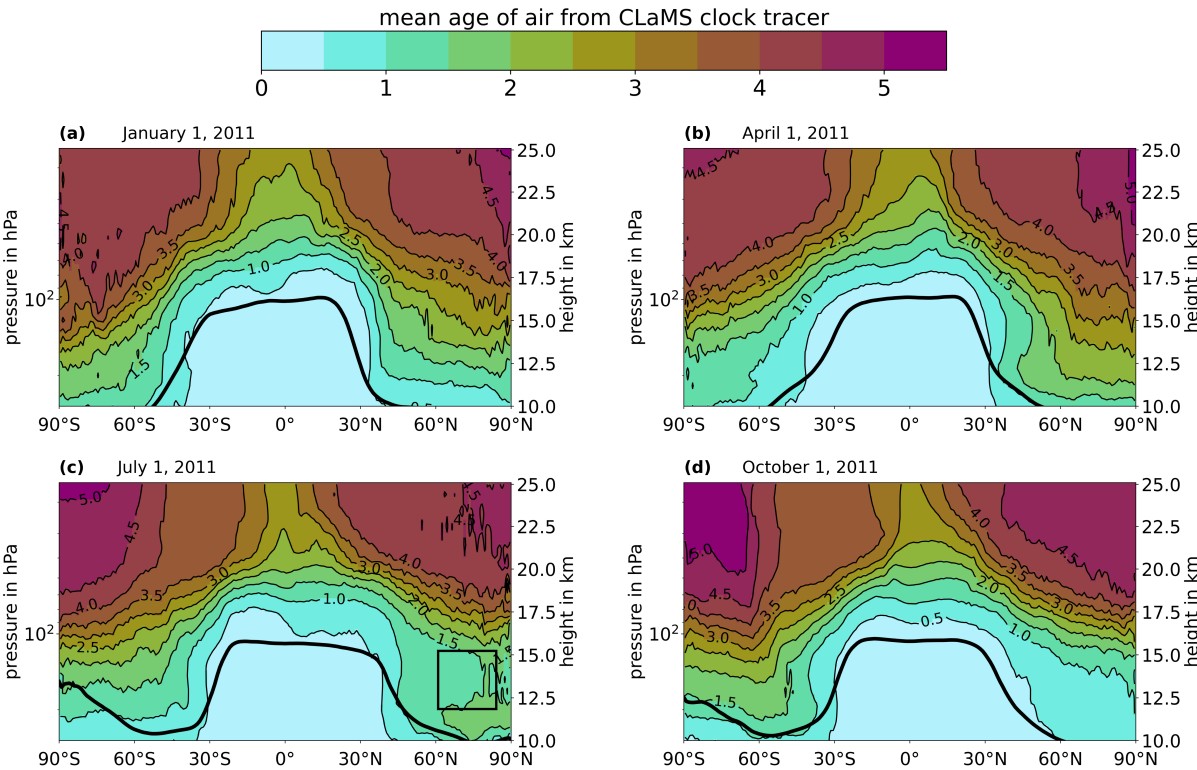

**Figure 6.** Color-coding and contours: Zonally averaged AoA from the CLaMS clock tracer in years at the four considered days in 2011. Thick black lines: Position of the zonally averaged tropopause according to ERA5 reanalysis for the four days (lapse rate tropopause following WMO (1957)). Black box in (c): Northward intrusion of young air masses

the clock tracer. The pseudo-measurements were created by adding normally distributed random noise to the mixing ratios of the six trace gases for all air parcels on the four considered days. The expected measurement uncertainty in Tab. 2 was used as the standard deviation of the noise distribution for each trace gas. In the next step, AoA and its total uncertainty was interpolated from the respective lookup table for each of these pseudo-measurements. Thereafter, the weighted mean AoA and its difference to the clock tracer AoA was calculated for each model grid point. Finally, zonal mean fields and standard deviations were calculated.

Figure 8 shows the zonally averaged weighted mean AoA for the four dates under consideration. The thick black lines represent the zonally averaged tropopause, derived from the daily ERA5 reanalysis data (lapse rate tropopause following WMO (1957)). The white areas correspond to those regions, where the zonal mean AoA from the clock-tracer of the model in Fig. 6 is below one year (tropospheric air). A comparison between both figures reveals that the general shape of the AoA distribution of the model was accurately recreated from the pseudo-measurements. Even the northward intrusion of air with AoA below one and a half years into the layer of air with AoA between one and a half and two years at roughly 70° N and

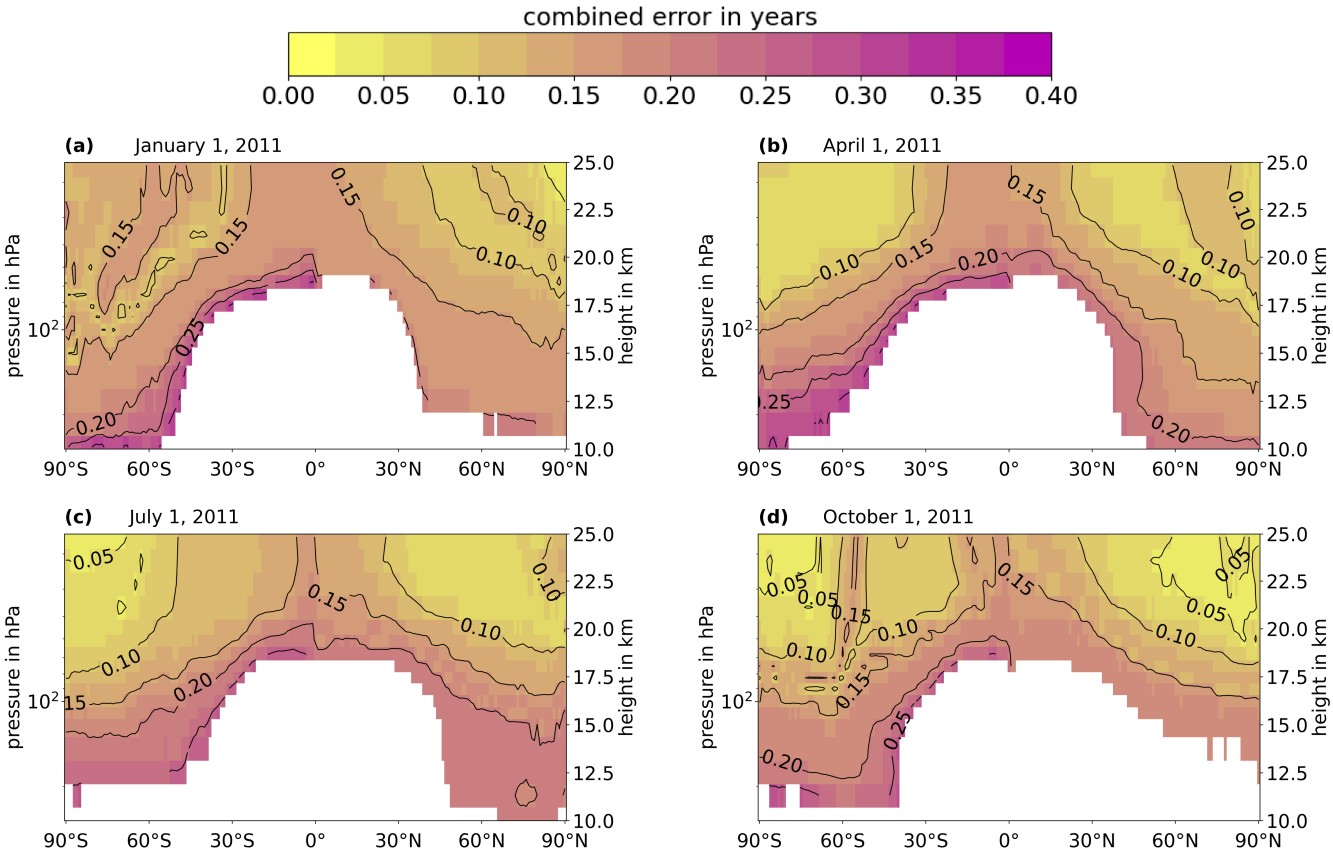

**Figure 7.** Combined total uncertainties in Fig. 5 and A3 mapped onto corresponding zonally averaged clock tracer AoA in Fig. 6. White areas correspond to regions with actual model AoA from clock tracer below one year (tropospheric air, not shown because of irrelevance and to avoid distraction)

14 km height in July is clearly present in the results of the correlation method (compare Figs. 6 (c) and 8 (c)). This intrusion of younger air is likely caused by outflow of the Asian summer monsoon (Ploeger and Birner, 2016; Krause et al., 2018).

Figure 9 shows the zonally averaged differences between the pseudo-measurement-derived AoA and the clock tracer AoA for the four days. The color-coded areas represent the absolute difference in years, and the contours represent the relative difference in percent. The white areas are identical to those in Fig. 8. At all four days, the absolute difference stays well below $\pm 0.3$ years in almost every region and does not surpass $\pm 0.5$ years anywhere. The absolute difference reaches its highest values of around 0.3 years at the upper end of the height-scale. These comparatively high values are most likely the result of high AoA values near the end of the lookup tables, for which the uncertainties for the individual tracers, and therefore the combined uncertainty of the weighted mean as well, tend to increase (compare Fig. 8 with Figs. 5 and A3). However, given that AoA values in these areas are particularly high, the relative difference between both types of AoA does not increase all that much

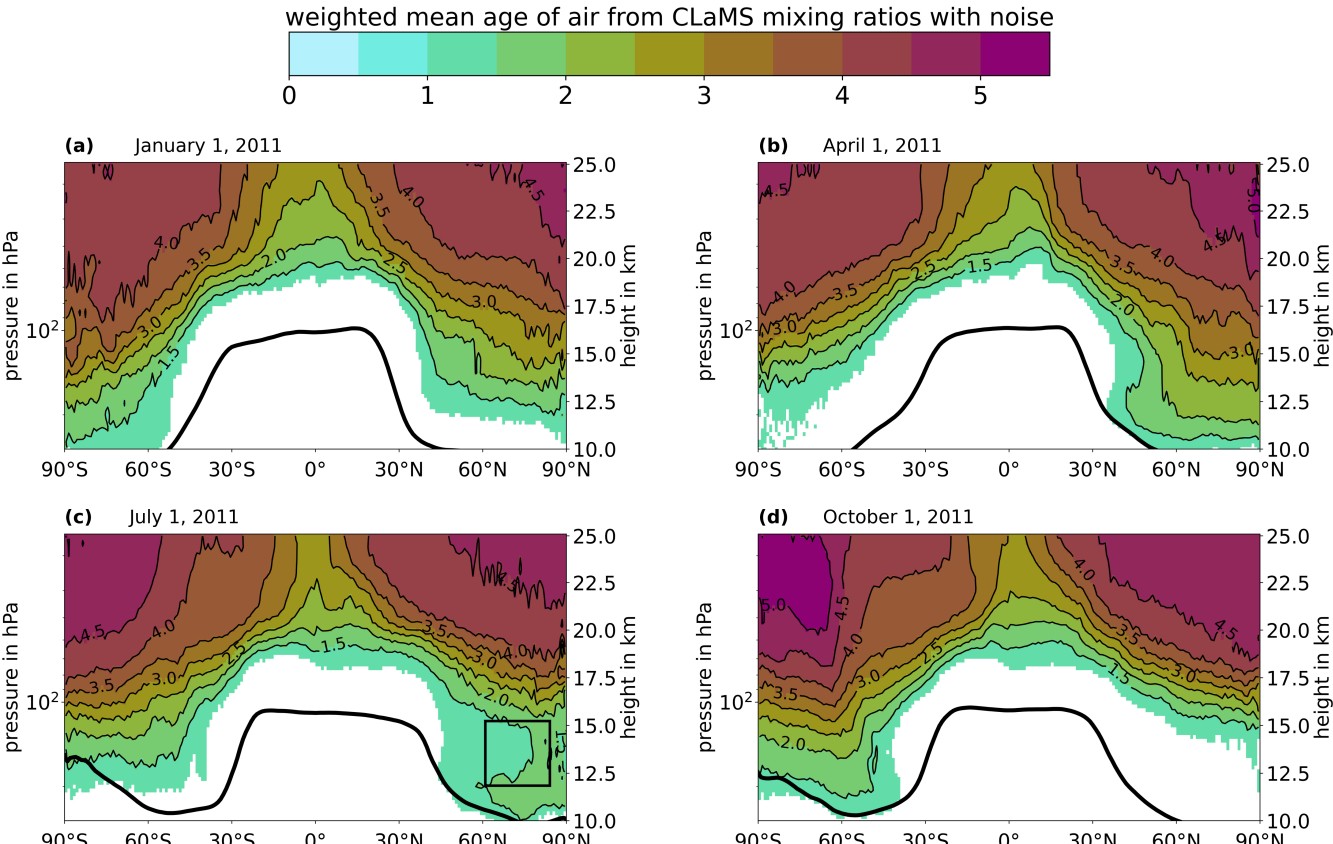

**Figure 8.** Zonally averaged weighted mean AoA calculated from the lookup tables for the six trace gases and synthetic measurements created with CLaMS. Thick black lines: Position of the zonally averaged tropopause at respective day according to ERA5 reanalysis (lapse rate tropopause following WMO (1957)). White areas correspond to regions with actual model AoA from clock tracer below one year (tropospheric air, not shown because of irrelevance and to avoid distraction). Black box in (c): Northward intrusion of young air masses

and stays below 10 % . The relative differences reach their highest values in the region near the tropopause, where AoA values are low and flat gradients inside the lookup tables cause the increased uncertainties (compare Figs. 4, A2, 5, A3 and 8).

Figure 10 presents the standard deviations in the zonal direction of the differences between pseudo-measurement and clock tracer AoA. These standard deviations quantify the spread of AoA differences in the zonal direction and can be used to estimate the uncertainty of AoA derived from individual measurements at different longitudes for any latitude. If the zonally averaged differences in AoA shown in Fig. 9 were uniformly zero, the standard deviations in Fig. 10 could directly represent these uncertainties. In such a scenario, Fig. 10 would closely resemble the initial uncertainty estimate in Fig. 7. However, as the differences in Fig. 9 are not uniformly zero, the two figures differ slightly. Specifically, the values in Fig. 9 are higher near the tropopause and towards the top of the height scale across all four days. To estimate the total uncertainty of AoA derived from individual measurements, the zonally averaged differences in AoA and the standard deviations in Fig. 10 must be combined as

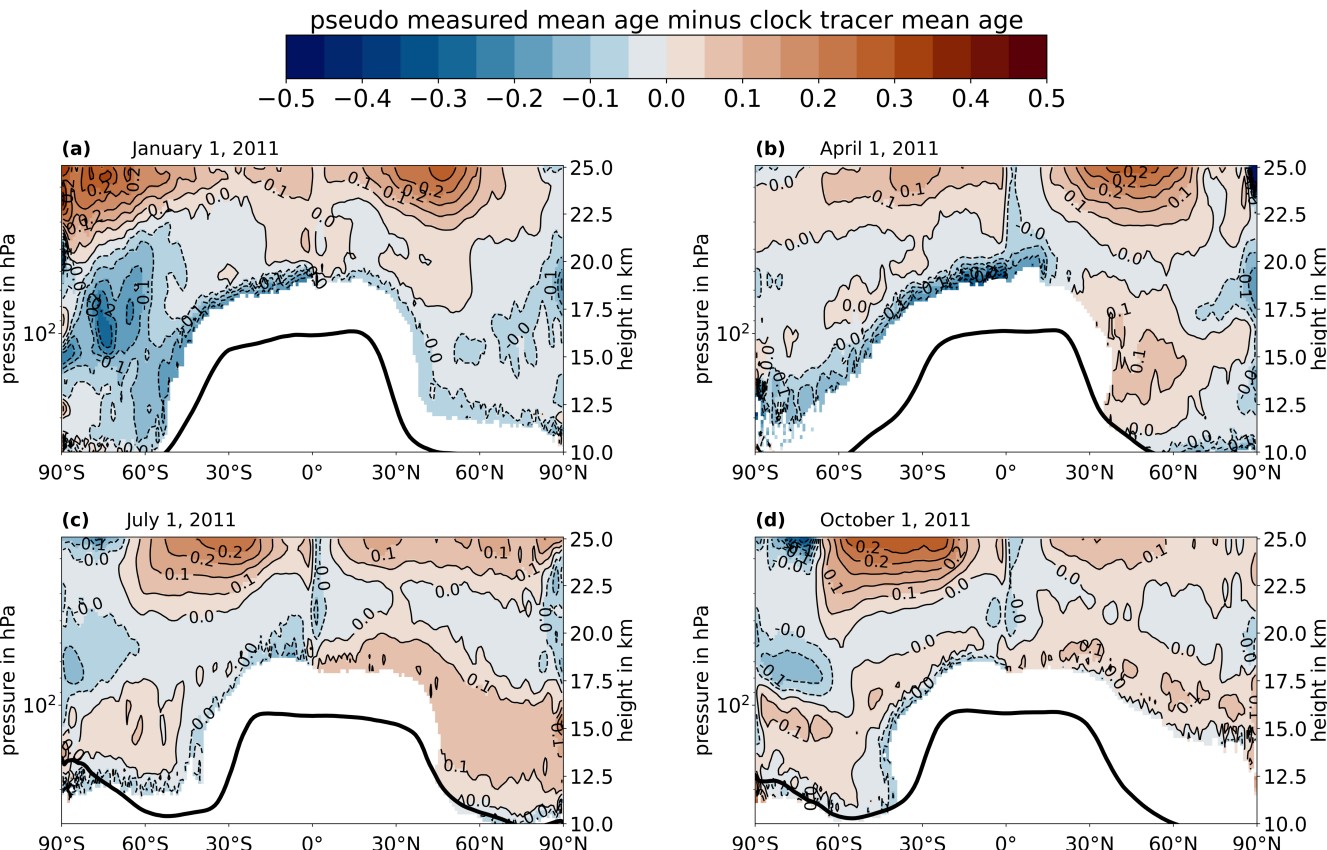

**Figure 9.** Zonally averaged difference between weighted mean AoA (from synthetic measurements with lookup tables) and actual CLaMS AoA (from clock tracer) in latitude-altitude-plane for the four days in 2011 (Color-coding: Absolute difference in years. Contours: Relative difference in percent.). Thick black lines: Position of the zonally averaged tropopause at respective day according to ERA5 reanalysis (lapse rate tropopause following WMO (1957)). White areas correspond to regions with actual model AoA from clock tracer below one year (tropospheric air, not shown because of irrelevance and to avoid distraction)

a Root Mean Square Error (RMSE). This calculation yields an average uncertainty of approximately 0.3 years for AoA at any latitude and longitude across the four days. A graphical depiction of the expected uncertainty for AoA derived from individual measurements is provided in Fig. S29 in the supplementary material.

## 3.4 Application to GLORIA balloon data

Figure 11 shows the mean AoA-profiles derived for the two GLORIA-B balloon flights as described in Sect. 2.3. The solid
blue lines ("GLORIA-B standard method") represent AoA calculated from observed $SF_6$ mixing ratios with the standard convolution method, as described in Garny et al. (2024b), and the subsequent correction for $SF_6$-depleted air from the mesosphere introduced by Garny et al. (2024a).

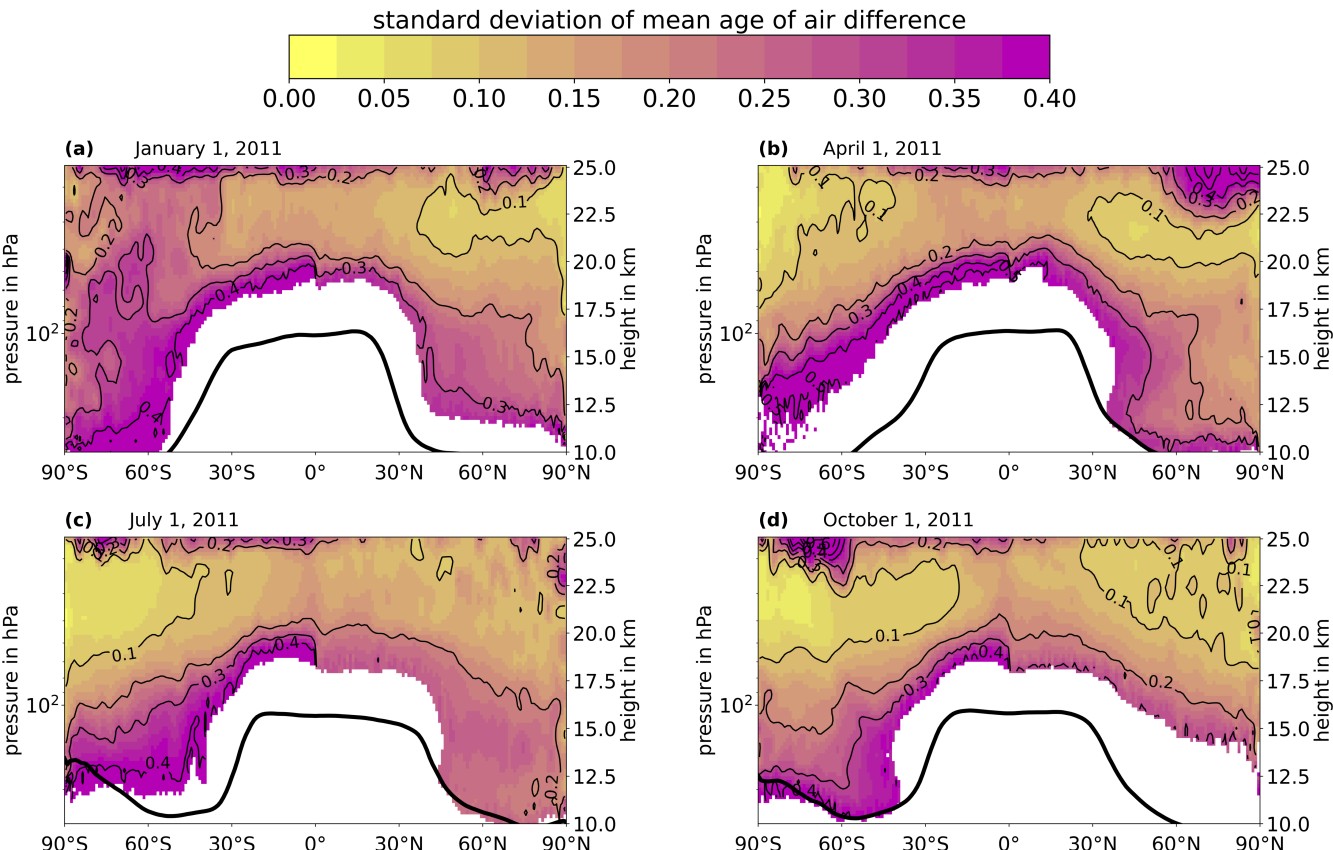

**Figure 10.** Standard deviations of differences between weighted mean AoA (from synthetic measurements with lookup tables) and actual CLaMS AoA (from clock tracer) in zonal direction for the four days in 2011. Color-coding: Standard deviations of absolute difference. Contours: Standard deviations of relative difference in percent. Note that the differences (absolute, relative) are calculated before the standard deviations. Thick black lines: Position of the zonally averaged tropopause at respective day according to ERA5 reanalysis (lapse rate tropopause following WMO (1957)). White areas correspond to regions with actual model AoA from clock-tracer below one year (tropospheric air, not shown because of irrelevance and to avoid distraction)

In the standard convolution method, the mean age of air spectrum is modeled using an inverse Gaussian distribution. This distribution is fully defined by its first two moments: the mean (AoA) and the width ($\Delta$). Alternatively, the inverse Gaussian can be specified using AoA and the ratio of its first two moments ($rom = \frac{\text{AoA}}{\Delta}$). Importantly, the ratio of moments remains relatively stable over time at a given location and can be determined from model-based or measurement-based lookup tables. Once the ratio of moments is known, the inverse Gaussian can be constructed for a range of AoA values. The method involves convolving the tropospheric $SF_6$ time series with the generated inverse Gaussians. The AoA value corresponding to the convolution result that best matches an observed $SF_6$ mixing ratio is then selected as the final estimate. Due to the unimodal nature of the inverse Gaussian distribution, the standard convolution method cannot resolve additional modes in the mean age of air spectrum that

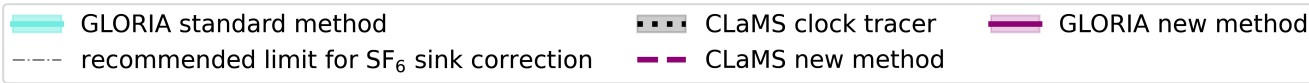

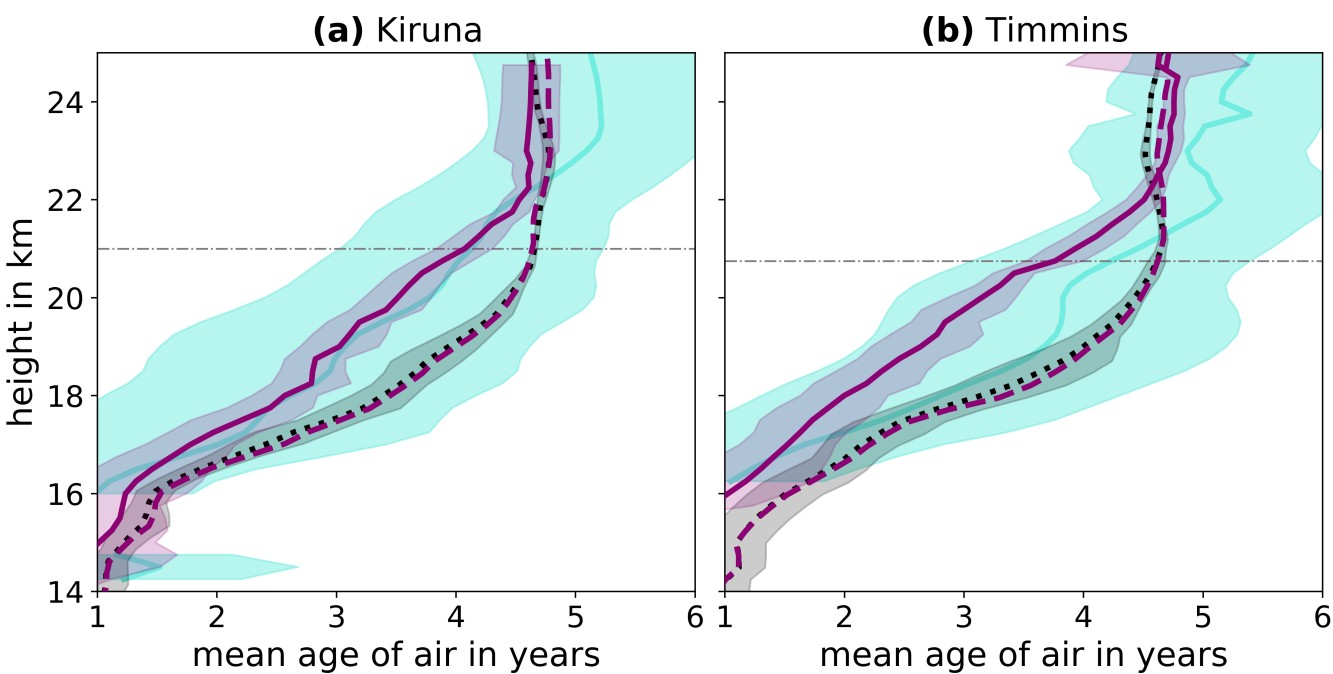

**Figure 11.** Mean AoA/height-profiles along measurement locations during balloon launches of GLORIA-B from Kiruna, Sweden (a) and Timmins, Canada (b). Blue lines: AoA with standard method (Garny et al., 2024b) and sink-correction (Garny et al., 2024a) from $SF_6$ observations. Blue shadings: Uncertainty ranges of blue lines from uncertainty of mean $SF_6$/height-profiles. Solid magenta lines: AoA with new correlation-based method (see Eq. 1) from observations of selected tracers (see Tab. 2) and lookup tables created from CLaMS (see Figs. S15 and S16 in supplement) with actual measurement uncertainties instead of estimated ones (see Tab. 2). Magenta shadings: uncertainty ranges of magenta lines (see Eq. 2). Dashed magenta lines: Same as solid magenta lines, but from CLaMS results interpolated on measurement locations. Dashed black lines: Actual CLaMS AoA from modeled clock tracer. Black shadings: One standard deviations around the respective mean values that constitute dashed black lines. Gray horizontal dash-dotted lines: Recommended height limit for $SF_6$ sink correction at which uncorrected AoA from standard method exceeds five years (Garny et al., 2024b).

may arise from the seasonality of the BDC. The AoA correction for $SF_6$-depleted air from the mesosphere is derived using a least-squares fit of an exponential function to global model results, establishing a relationship between uncorrected and actual AoA. While the uncertainty of this correction method increases with AoA, it is optimized for AoA values up to 5 years. Despite this limitation, the method continues to enhance the accuracy of results even for AoA values exceeding 5 years.

The horizontal gray dash-dotted lines mark the limit where the correction method for mesospheric depletion becomes less accurate as the uncorrected AoA exceeds the stated limit of 5 years (Garny et al., 2024a).

The blue shaded areas around the "GLORIA-B standard method" AoA represent its uncertainty range (see Sect. 2.3). The solid magenta lines ("GLORIA-B new method") represent the weighted mean AoA calculated from the observed mixing ratios of the six selected trace gases listed in Tab. 2 via the proposed correlation method (see Eq. 1). The required lookup tables to identify each of the six trace gases with corresponding AoA and uncertainty were created from the CLaMS results for the northern hemisphere and the day of the respective balloon launch (Kiruna: 21/08/2021, Timmins: 23/08/2022). These lookup tables were created in exactly the same way as the ones for the four days in 2011 discussed in Sect. 3.2. Illustrations similar to those in Fig. A2 for the lookup tables at the two flight days can be found in Figs. S15 and S16 in the supplement. However, instead of the estimated measurement uncertainties of the CAIRT instrument in Tab. 2, the actual uncertainties of the GLORIA-B instrument for the measurement and temporal averaging of each trace gas were used to estimate the measurement part of the total AoA uncertainty. The magenta shading around the "GLORIA-B new method" AoA represents its uncertainty according to Eq. 2. A comparable AoA profile has been calculated from the CLaMS model mixing ratios of the six species interpolated on the measurement locations (dashed magenta lines), and these are shown together with the actual model age profiles from the model clock-tracer (dashed black lines). The black shaded areas around the "CLaMS clock tracer" AoA represent the standard deviations of the actual model AoA along the recreated points of observation.

The comparison of the "CLaMS clock tracer" AoA with the "CLaMS new method" AoA shows that the new correlation method produces reliable results for both flights, such that inferred mean age agrees very well with the model clock-tracer age. For either flight, the standard deviation of the actual CLaMS AoA from the clock-tracer does not exceed half a year and the difference between the two model AoA profiles is even smaller at most height levels. Hence, the new correlation method passes the proof-of-concept in the consistent model environment as well as when applied to single balloon profiles.

The values of "GLORIA new method" fall entirely within the uncertainty range of "GLORIA standard method" AoA for both flights, but the reverse is true only for the flight from Kiruna below roughly $22\,\mathrm{km}$. Further, the two profiles for the flight from Kiruna seem to agree much better with each other than for the flight from Timmins. For the Timmins flight, on the other hand, "GLORIA standard method" is up to one year older than and lays outside the uncertainty range of "GLORIA new method" over nearly the entire height range. Given the extraordinarily high uncertainty of "GLORIA-B standard method", however, any conclusions drawn from the corresponding values are hardly meaningful. Note that the aforementioned high uncertainty in AoA is the result of the high measurement uncertainty of $SF_6$. A possible explanation for the higher difference between the two profiles for the Timmins flight could be that the air masses scanned by GLORIA-B were influenced by $SF_6$-depleted air from the mesosphere to a higher degree compared to the Kiruna flight, and wasn't fully compensated by the applied $SF_6$ sink correction. At least for the results in the last roughly four kilometers of the height scale, which are above the recommended limit for the application of the $SF_6$ sink correction, this explanation seems plausible. Another possible explanation would be some issue with the instrument during the Timmins flight that led to the retrieval of systematically too low $SF_6$ mixing ratios. There is, however, no indication of what could have caused such an issue thus far.

Overall, a clear improvement in the new age calculation method using tracer correlations is the substantial reduction in AoA uncertainty — from several years to around half a year — across nearly the entire height range for both flights. The two profiles of "GLORIA-B new method" are also in good agreement with the recently published AoA profiles by Schuck et al. (2024b).

In their study, Schuck et al. (2024b) calculated AoA values using cryogenic whole-air samples of $SF_6$ and $CO_2$ collected in Kiruna, Sweden, during August 2021. Similar to the AoA profiles of "GLORIA-B new method" the results reported by Schuck et al. (2024b) exhibit an increase in AoA with altitude, reaching approximately five years at around 22 km. Beyond this altitude, the AoA values remain approximately constant at their maximum of five years. There are good reasons to believe that the utilized model-based lookup tables do reflect the actual atmospheric conditions well enough to justify their application to the GLORIA-B measurements. These reasons will be discussed in the next section. On top of that, an alternative to the model in the form of global satellite measurements as the foundation of the lookup tables will also be discussed in the next section.

Comparison between model and measurement-based age of air in Fig. 11 further shows that the results of the new correlation-based method applied to the two different sets of measurements by GLORIA-B are consistently higher than the model results up to around 22 km. Hence, the stratospheric BDC in the ERA5 reanalysis-driven CLaMS simulation appears to be significantly slower compared to the observations. This result is in agreement with a generally slower BDC in ERA5 compared to other reanalysis and observations as found by (Ploeger et al., 2021a; Laube et al., 2020). It should be noted that this slow bias of the ERA5 circulation in the comparison presented here is independent of the age calculation method used.

## 3.5 AoA from synthetic measurements with zonally averaged correlations

For a given satellite instrument it can be more reliable to derive trace gas mixing ratios for the zonal mean than for individual profiles (e.g. Stiller et al., 2012). Therefore, we apply the new correlation-based method to calculated AoA for zonal mean data, in the following, and further discuss its applicability in Sect. 4. For each of the four considered dates in 2011 (first of January, April, July and October), zonal mean fields for trace gas mixing ratios and clock tracer AoA values have been calculated from the CLaMS model data. Lookup tables to obtain AoA from mixing ratios were then created from these zonal mean fields in the same way as for the 3-D data in Sect. 3.2 and as described in Sect. 2.2 (see Figs. 4 and A2). Just like in Sect. 3.2, one lookup table for each trace gas, day, and hemisphere was created, making 48 lookup tables in total. These lookup tables from the zonal mean fields were then used to calculate weighted mean AoA from the previously calculated 3-D (zonally-dependent) synthetic measurements of the six trace gases using Eq. 1. These weighted mean AoA values were then evaluated by comparison to the actual AoA of the model obtained from the clock tracer. The evaluation was done in the same way as before for the weighted mean AoA calculated from the lookup tables created from the full 3-D data set (see Sect. 3.3).

Figure 12 shows exemplarily for 1 July 2011 the different steps in the calculation and evaluation of the weighted mean AoA for the calculation based on zonal mean data. Figure 12a shows one example from the 48 lookup tables created from the zonal mean fields: the one for CFC-12 in the southern hemisphere. The equivalent lookup table from the original CLaMS data sets can be seen in Fig. 4c. Note that the number of mixing ratio bins was reduced from 150 to 40 for the lookup tables from the zonal mean fields. A smaller number of bins was necessary because of the significant reduction of data points caused by the zonal averaging. Graphical depictions of the lookup tables for the remaining cases can be found in Figs. S17 to S28 of the supplement to this article.

Figure 12b shows the individual total uncertainties of AoA from the lookup tables of the six trace gases in the southern hemisphere as well as their combined uncertainty calculated with Eq. 2 (see Figs. S30 and S31 in the supplement for northern

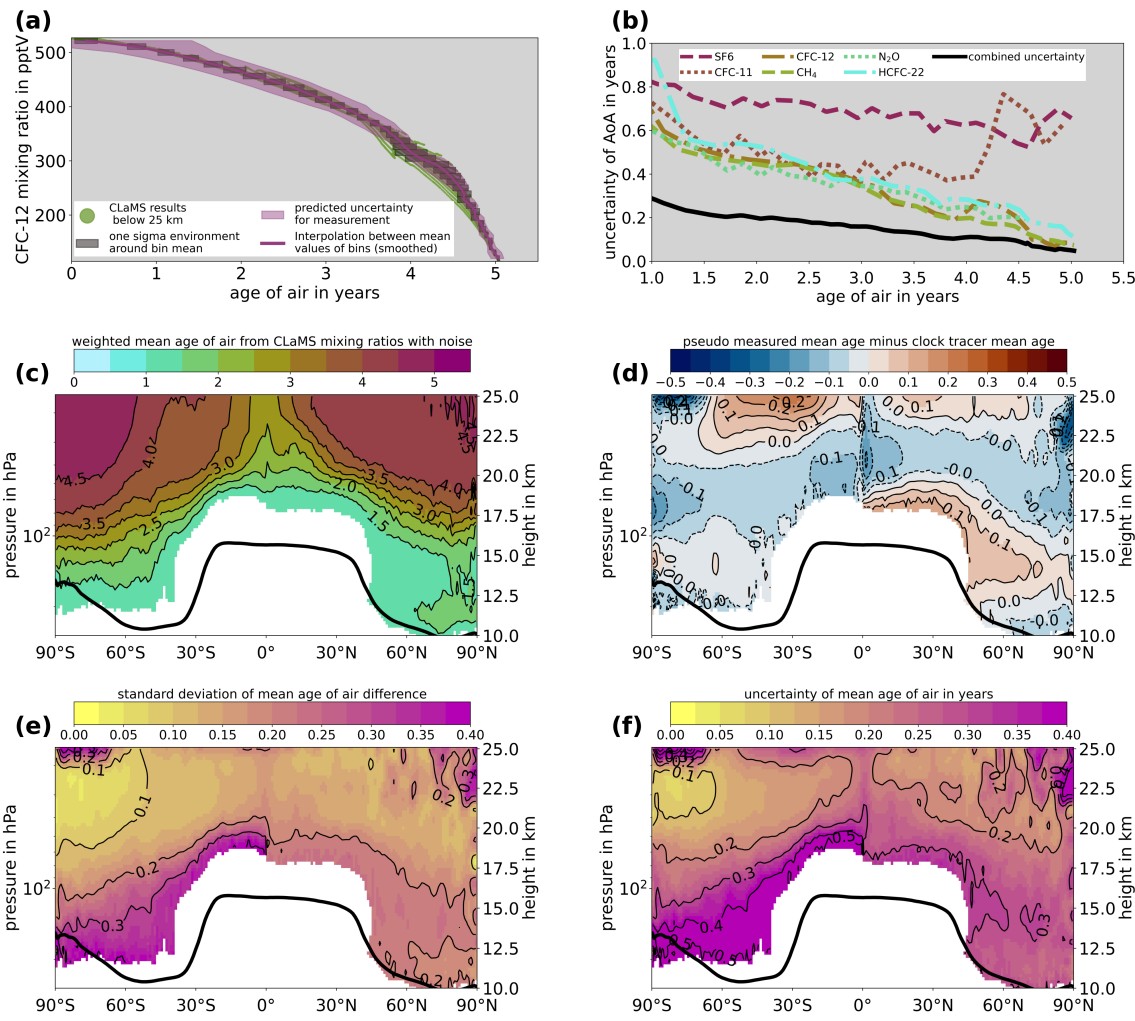

**Figure 12.** Selection of plots exemplifying the analysis in Sec. 3.3 for lookup tables created from zonally averaged CLaMS results. All plots refer to 1 July 2011. Bold black line in (c) to (f) indicates zonally averaged tropopause according to ERA5 reanalysis (lapse rate tropopause following WMO (1957)). **(a)** Creation of lookup table for zonal mean AoA from zonal mean CFC-12 in southern hemisphere (see Figs. 4 and A2). **(b)** Dashed/dotted/dash-dotted colored lines: Total uncertainty of each tracer in the southern hemisphere. Solid black line: Combined total uncertainty for all tracers according to Eq. 2 (see Figs. 5 and A3). **(c)** Zonally averaged weighted mean AoA from synthetic measurements with lookup tables (see Fig. 8). **(d)** Zonally averaged difference between weighted mean AoA from synthetic measurements and CLaMS clock tracer (see Fig. 9). **(e)** Standard deviations of zonally averaged differences between weighted mean and clock tracer AoA (see Fig. 10). Note that the differences are calculated before the standard deviations. **(f)** Estimation of uncertainty of weighted mean AoA from lookup tables along all longitudes (values in (d) and (e) added together as a RMSE).

hemisphere and remaining days). This figure corresponds to Fig. 5c which shows the respective uncertainties from the lookup tables created from the original CLaMS data sets.

The lookup tables from the zonal mean fields were applied to the 3-D synthetic trace gas measurements to create a weighted mean AoA for every model grid point. Thereafter, the differences between these weighted mean AoA values and the corresponding actual AoA values from the clock-tracer of the model were calculated, and the mean values and standard deviations of these differences in zonal direction constructed. The weighted mean AoA values were zonally averaged as well, the result of which is shown in Fig. 12c (see Fig. S32 in the supplement for remaining days). The AoA distribution in Fig. 12c is essentially the same as the corresponding one from the full 3-D CLaMS data set in Fig. 6.

Fig. 12d shows the zonally averaged differences between the weighted mean AoA and the actual AoA of the model (see Fig. S33 in the supplement for remaining days). The differences displayed in Fig. 12c are similar in size and distribution to the corresponding ones from the non-averaged CLaMS data sets shown in Fig. 8c. The average difference is slightly higher in the case of the lookup tables from zonal mean data, at 0.06 years, than in the case of the lookup tables from the non-averaged , where it is 0.02 years, but these differences are very minor.

Fig. 12e shows the standard deviations in zonal direction of the difference between actual model and weighted mean AoA (see Fig. S34 in the supplement for remaining days). These standard deviations are also remarkably similar to the corresponding ones of the full zonally-dependent calculation (Fig. 10).

Fig. 12f is the result of Fig. 12d and Fig. 12e added together as a RMSE (color-coding, not contours) and represents an estimation for the zonally averaged uncertainty of the weighted mean AoA (see Fig. S35 in the supplement for the remaining days). In other words, if the uncertainties of the weighted mean AoA values (see Eq. 2) were zonally averaged, the results should closely resemble the color-coded values in Fig. 12f. Figure S29 in the supplement is the correspondence to Fig. 12f for the lookup tables from the non-averaged CLaMS data. Since there are also no major differences between the two cases in this final uncertainty estimation, it can be concluded that estimation of the relation between trace gas mixing ratio and mean age from zonal mean data yields just as reliable results than the estimation from the full 3-D data. This conclusion can be drawn for all four days considered (see the respective figures in this article and supplement).

## 4 Discussion

### 4.1 Towards an application on satellite measurements

The proposed method to derive AoA requires knowledge of its functional relations with respect to mixing ratios of certain trace gases. In a model environment, where the true AoA values are known, these functional relations can be easily established. It is, however, unclear how well such model-based relations might reflect the actual conditions in the real atmosphere. Therefore, it is preferable to establish the required functional relations directly from the satellite measurements itself rather than from model results.

For that purpose, we suggest to first calculate AoA from $SF_6$ with well-established standard methods like the convolution method proposed by Garny et al. (2024b) in combination with the $SF_6$ sink correction by Garny et al. (2024a), as a first guess.

Given the difficulty in retrieving $SF_6$ from satellite measurements, this first guess AoA would be afflicted with significant uncertainties that make interpretations of individual values impossible. In order to reduce the measurement noise, the first guess AoA is likely possible only for data averaged over larger areas (e.g. zonal mean data). The required lookup tables could be calculated from the averaged first guess AoA and likewise averaged mixing ratios with a method similar to the one described in Sect. 2.2. The thus established lookup tables could then be used to calculate weighted mean AoA at the individual points of observation inside the respective areas. The necessary size of these areas (e.g. the length of longitude ranges for zonal averaging) will depend on the amplitude of the noise in future satellite measurements of $SF_6$ and needs to be specifically defined for the satellite instrument considered.

In case of a large $SF_6$ measurement uncertainty, the first guess AoA would have to be calculated for averages over the entire longitude range (zonal mean data) to reach a sufficient reduction in noise to enable the construction of the required lookup tables. On the basis of observations by MIPAS, Stiller et al. (2012) have demonstrated that reliable zonal mean AoA values can in fact be derived from existing satellite measurements of $SF_6$. The results in section 3.5 suggest that it should be possible to derive AoA with an uncertainty of just a few percent even from zonal mean trends in most of the lower stratosphere throughout the year. For future satellite measurements with further reduced uncertainty in the $SF_6$ retrieval it is to be expected that the areas for the necessary averaging can be further reduced and that thereby the accuracy of the estimated AoA can be further increased.

## 4.2 Further improvements

This study presents a proof of concept that AoA could potentially be derived more accurately and with higher spatial and temporal resolution from satellite measurements, if trace gases other than $SF_6$ are taken into account as well. In order to make the general idea of deriving AoA from correlations of trace gases with AoA easier to understand, a relatively simple method was developed in this study. Several measures to improve this simple method could be worth exploring before employing it on actual satellite observations. One potential way the method might be improved could be a separation of data points into narrower latitude bands than just the two hemispheres considered here. It might, for instance, be preferable to consider the tropical latitudes separately, since these constitute a unique region in terms of dynamics that is characterized by fast upwelling. A separation into different altitude-bands might also be a valuable improvement of the method.

Another way the presented method might be improved is by including a larger number of trace gas species. Additional trace gas species to the ones considered in this study with long stratospheric lifetimes might also show sufficiently compact correlations with AoA in the lower stratosphere. Including more trace gases with such compact correlations would be useful to further reduce the uncertainty of AoA derived with the presented method. However, all these species need to be retrieved from satellite observations with reasonably low uncertainty. Potential trace gases with the required properties could include certain species of chlorofluorocarbons (CFCs), hydrochlorofluorocarbons (HCFCs), hydrofluorocarbons (HFCs) and Perfluorocarbons (PFCs).

## 4.3 Model and method uncertainties

This study should, first and foremost, be viewed as a proof of concept for the proposed method to calculate AoA. Such a proof of concept can only be demonstrated within the perfectly self-consistent environment of a model, where the "true" values of the desired quantity are already known. Since no model is a perfect representation of the actual atmosphere, the lookup tables required for applying the proposed method should ideally be constructed directly from measurements, as laid out in Sect. 4.1, rather than from model results. Due to the lack of suitable measurement data, we had to establish the lookup tables required for applying the method to the GLORIA-B data using the available model results. We are confident that the CLaMS-based lookup tables used on the GLORIA-B data represent the actual atmospheric conditions reasonably well and find the AoA profiles derived from these lookup tables to be more plausible than the corresponding ones derived with the standard method (see Fig. 11). Our confidence in the CLaMS-based lookup tables stems from the following three reasons:

1. The reliability of the CLaMS model for the utilized trace gases has been demonstrated in previous studies (see, e.g., Laube et al. (2020) for CFC-11, CFC-12, and HCFC-22, Konopka et al. (2004) for $CH_4$, and Pommrich et al. (2014) for $N_2O$).

2. There is excellent agreement between the AoA profiles reported by Schuck et al. (2024b) and those derived using the proposed method from the GLORIA-B measurements at Kiruna (see Fig. 11). The balloon-borne air samples collected by
Schuck et al. (2024b) should closely resemble the air masses observed by GLORIA-B during its first flight, as both balloon flights took place in August 2021 in Kiruna, Sweden. The fact that Schuck et al. (2024b) obtained similar results for comparable air masses using a different calculation method strongly supports our expectation that the utilized CLaMS-based lookup tables reflect the conditions of the actual atmosphere reasonably well.

3. The upper boundary region, arguably the biggest source of bias in the simulation as a whole, seems to have only limited impact on the model tracer mixing ratios in the considered altitude region (i.e., below $25\,km$). Figure A4 in the appendix shows the temporally averaged zonal mean $SF_6$ distribution of the model results up to the upper boundary over two different three-month periods. Fig. 11a shows the average distribution from December 2010 to February 2011, representing winter in the Northern Hemisphere, and Fig. 11b shows the average distribution from June 2011 to August 2011, representing winter in the Southern Hemisphere. The winter periods were chosen because they represent the time when the downward transport of air from higher altitudes by the deep branch of the BDC is strongest in the respective hemisphere. $SF_6$ was selected because, in relative terms, it reaches the upper boundary in higher amounts than all other tracers (due to the absence of a stratospheric sink). Out of all model results, the influence of the upper boundary is therefore expected to be strongest for $SF_6$ during winter in the respective hemisphere. Figure 11 suggests that the downward transport 11, the downward transport of $SF_6$ depleted air from the upper boundary only seems to have an affect below $25\,km$ at high latitudes in the winter hemisphere. This limited influence of the upper boundary on model results below $25\,km$ could partly be related to the slow bias of the stratospheric ERA5 circulation, as pointed out in sect. 3.4. This

slow bias, however, effects the six long-lived trace gases in the same way as the model clock tracer and therefore can't
be responsible for any biases the model-based lookup tables might have.

While these considerations give us confidence in the plausibility of the CLaMS-based lookup tables, it is important to emphasize that a thorough analysis would be necessary to fully assess the influence of the upper boundary on each of the six trace gases specifically, and to evaluate the extent to which the derived lookup tables reflect the conditions of the real atmosphere. Such an analysis would ideally involve the use of several atmospheric models and/or reanalysis datasets to better
quantify potential biases and uncertainties. Additionally, the influence of uncertainties in the kinetic model parameters, such as photochemical data and model-calculated loss rates, on AoA cannot be ruled out. A thorough analysis of the influence of these uncertainties on AoA would also be needed. However, conducting such extensive analyses is beyond the scope of this study.

## 5   Conclusion

We have introduced a new method to derive stratospheric AoA from tracer mixing ratios in order to reduce the high uncertainty
that arises when stratospheric AoA is derived from remote sensing observations using a common standard method. The standard method to calculate AoA from trace gas observations is only applicable for trace gases that approximate an ideal clock tracer sufficiently well. The only approximate clock tracer that has been retrieved from satellite observations within a sufficiently meaningful uncertainty range is $SF_6$ (Stiller et al., 2008). The uncertainty of retrieved $SF_6$ is, however, still significant. Standard methods (Garny et al., 2024b) have therefore, until now only been applied on zonal averaged satellite measurements of $SF_6$. The
new correlation-based method to calculate AoA introduced in this study is an attempt to make use of the full spatial resolution of satellite observations and to avoid the necessity to restrict to zonal mean data. With this new method, AoA can be derived from any trace gas that holds a compact correlation with AoA in the stratosphere and not just from approximate clock tracers like $SF_6$. Using other trace gas species in addition to $SF_6$ to derive AoA has the advantage that the uncertainty associated with the AoA estimate gets reduced with every new species with sufficient high accuracy added. Although the methods to derive
AoA proposed by Leedham Elvidge et al. (2018) and Ray et al. (2024) also use other trace gases in addition to $SF_6$, they are not applicable on typical remote sensing measurements, because they relay on trace gases that cannot be retrieved without a significant uncertainty.

The applicability of the new method introduced in this study on atmospheric remote sensing observations has been demonstrated based on balloon-borne measurements. The application of the new method produced significantly lower AoA uncer-
tainties than the application of the standard method and the results for the new method fell entirely within the uncertainty range of the results for the standard method. Comparison of mean age from the balloon-borne measurements with ERA5 reanalysis-driven model simulations showed that the new correlation-based method provides mean age profiles with sufficiently reduced uncertainty compared to standard methods to truly enable constraining the model. Consequently, it could be shown that the stratospheric circulation in the ERA5 reanalysis data is slower than the actual stratospheric circulation in the real atmosphere.
Furthermore, it is still an open question whether or not - and if so, where exactly and to what degree - the BDC will accelerate due to the changing climate and the associated warming of the troposphere and cooling of the stratosphere. Age of air

datasets calculated from observations with the new correlation method could be used to validate simulated circulation trends. Furthermore, such datasets could be invaluable for studying exchange processes between the troposphere and the stratosphere. These studies have wide-ranging applications in atmospheric science, including but not limited to: understanding the transport of water vapor and other trace gases, investigating the impact of stratospheric ozone on climate, assessing the influence of tropospheric pollutants on stratospheric chemistry, and helping to better constrain emissions of various substances, including prohibited ones like CFCs.

It was demonstrated on the basis of model output from CLaMS that the new method produced reliable zonal average AoA distributions in the lower stratosphere. These retrieved age distributions even showed smaller-scale variations consistent with the actual AoA from the clock-tracer like a layer of relatively young AoA in higher altitudes in the Northern summer lowermost stratosphere caused by the outflow of the Asian summer monsoon. The analysis with CLaMS also showed the potential usefulness of the new method for deriving AoA on individual satellite observation points in the lower stratosphere throughout the year. The average uncertainty of AoA derived with the new method is expected to be around $0.3$ years at any day and location. The application of the new correlation-based method to derive AoA on future limb-imaging satellite observations such as CAIRT could therefore provide spatially-resolved global distributions of AoA in the lower stratosphere with significantly reduced uncertainty. Such datasets do not exist up to this point and are urgently needed to enhance our understanding of transport processes in the stratosphere and to study the effects of these processes by improving their representation in climate models.

## Appendix A

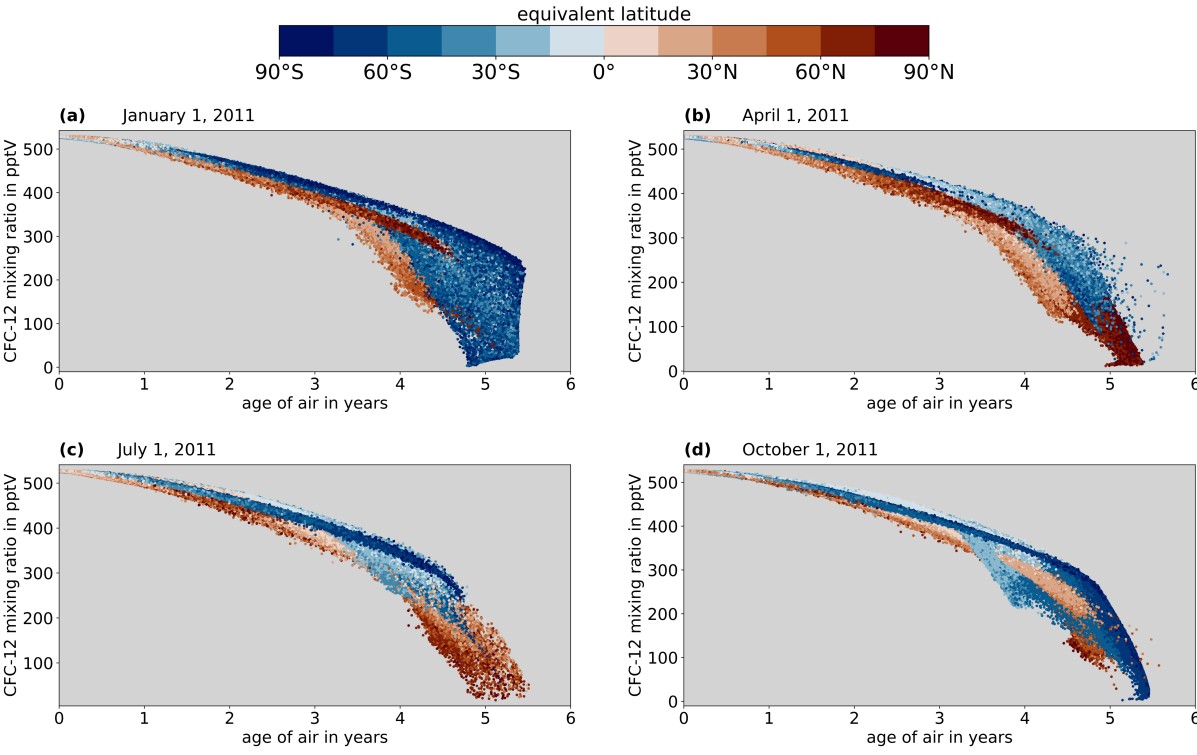

**Figure A1.** Same as Fig. 3, but for CFC-12 instead of SF6.

*Author contributions.*  FV performed the investigation, data analysis, data curation and visual representation; FP, JL and PP initiated the study, conceptualized the core research questions and contributed to writing the manuscript; PK developed the setup for the CLaMS simulation; JUG implemented the new trace gas species $SF_6$ and HCFC-22 into the CLaMS model; JL helped to develop the lower and upper boundary conditions of the six trace gases; PP and JU developed the new method for calculating AoA; ER contributed to developing the new method for calculating AoA by conceiving a significant improvement to the original idea; GS provided the MIPAS datasets for $SF_6$ and AoA that

were used to create the upper boundary condition of $SF_6$, and the initial mixing ratio distributions of $SF_6$ , $CH_4$ , $N_2O$ and HCFC-22 in the CLaMS simulation; JU, MH, GW and SJ performed the processing of the GLORIA-B data; GW and SJ have provided the mixing ratio profiles with uncertainties for the balloon launches of GLORIA-B. All Co-Authors contributed in discussions and finalizing the manuscript.

*Competing interests.*  At least two of the (co-)authors are a member of the editorial board of *Atmospheric Chemistry and Physics*.

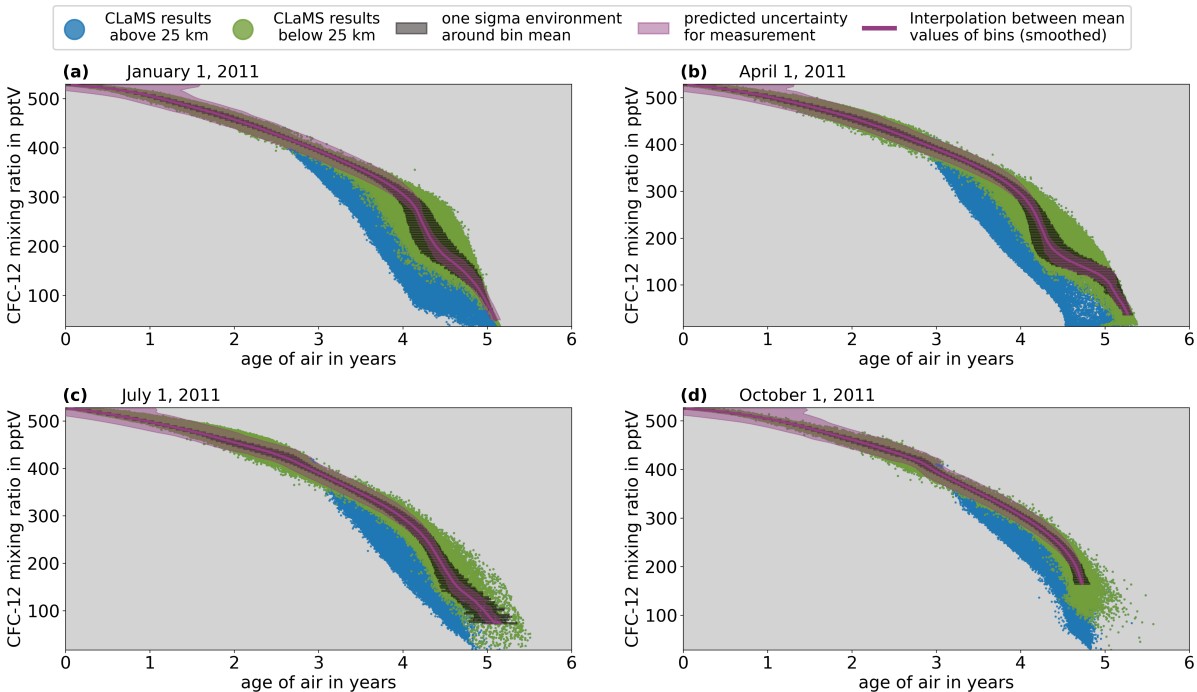

**Figure A2.** Same as Fig. 4, but for northern hemisphere.

*Code and data availability.* The CLaMS models are available in the Modular Earth Submodel System (MESSy) Git database. Detailed information is available at https://messy-interface.org/licence/application (Modular Earth Submodel System, 2024). ERA5 reanalysis data are available from the European Centre for Medium-range Weather Forecasts (https://apps.ecmwf.int/data-catalogues/era5/?class=ea, ECMWF, 2024). The CLaMS model data used for this paper may be requested from the corresponding author (f.ploeger@fz-juelich.de). The lookup tables created for this study are available online at: https://doi.org/10.5281/zenodo.14543944

*Acknowledgements.* The authors extend their deepest gratitude to the Centre National d'Études Spatiales (CNES) for their invaluable role in executing the balloon launches of the GLORIA-B instrument from Kiruna, Sweden, and Timmins, Canada. These operations were carried out under the auspices of the HEMERA research infrastructure, supported by the Horizon 2020 framework program of the European Union. We also wish to acknowledge the entire GLORIA team for their dedication and expertise, with special thanks to the technology institutes ITE at Forschungszentrum Jülich and the Institute for Data Processing and Electronics at the Karlsruhe Institute of Technology, whose efforts were instrumental in obtaining the measurement results utilized in this study. Our sincere thanks go to the Global Monitoring Laboratory of the National Oceanic and Atmospheric Administration (NOAA) for providing the essential mixing ratio time series that served as the lower boundary conditions in the CLaMS simulation. We also gratefully acknowledge the MIPAS team for supplying the $SF_6$ and AoA data sets, which were crucial for initializing the CLaMS simulation and defining the upper boundary condition for $SF_6$. We would like to express our appreciation to Nicole Thomas and Patrick Jöckel for their support and assistance in setting up the model simulations. Finally, we are deeply grateful for the computing time granted on the supercomputer JUWELS at the Jülich Supercomputing Centre (JSC), provided under

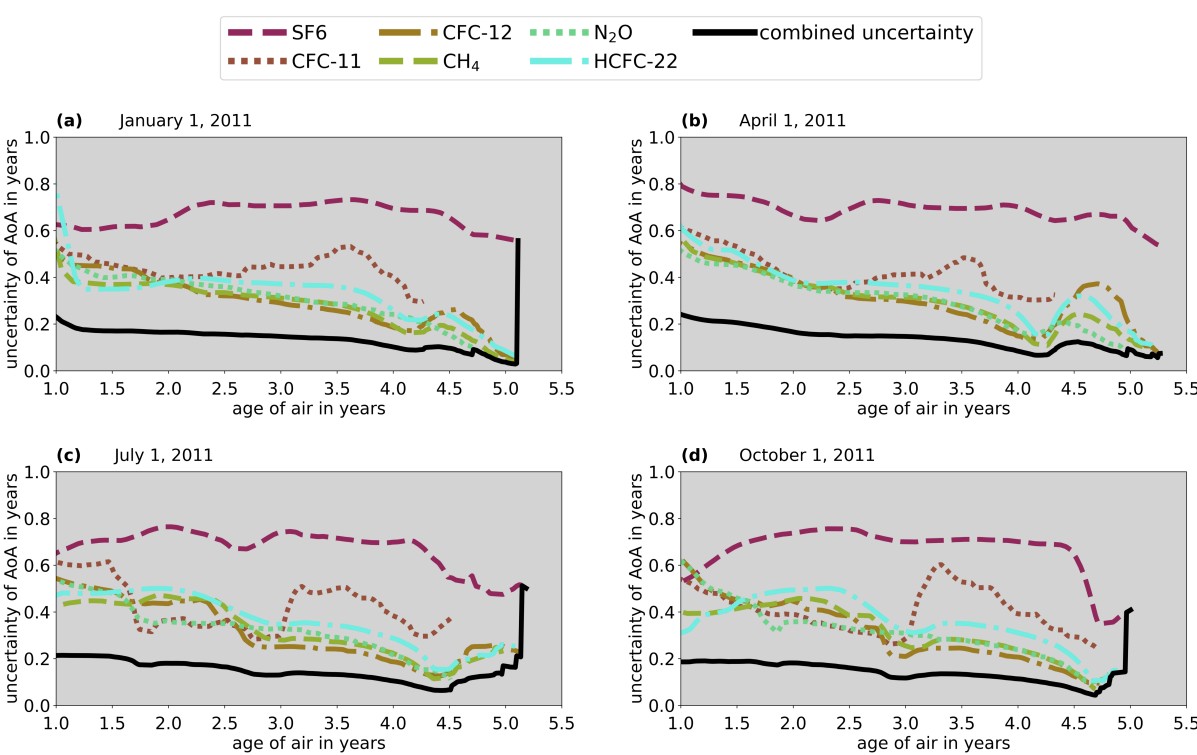

**Figure A3.** Same as Fig. 5 but for northern hemisphere

the VSR project ID CLAMS-ESM, which was vital for the completion of the CLaMS simulations. This study was funded by the Deutsche Forschungsgemeinschaft (DFG, German Research Foundation) – QUANTITEE – project ID 462476233 and supported by the European Space Agency (ESA) via the MAGIC-4AMPAC project (4000138370/22/NL/SD) and the CAIRTEX project (4000141036/23/NL/FF/ab).

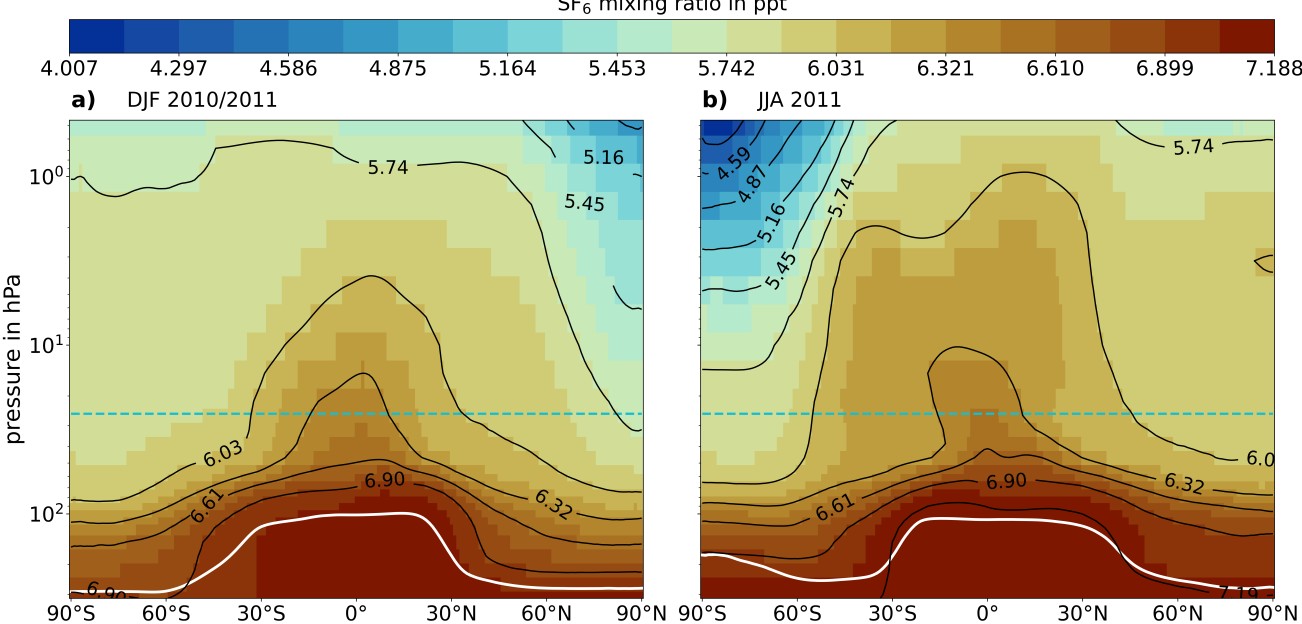

**Figure A4.** Zonal mean SF$_6$ mixing ratio from CLaMS results from tropoause to stratopause region averaged over a three month period. a): Averaged over December of 2010 and January and Feburary 2011. b) Averaged over June, July and August of 2011. Dashed blue line: Upper limit for applicability of proposed method to calculate AoA. White line: Zonal mean tropopause form ERA reanalysis averaged over respective three month period (lapse rate tropopause following WMO (1957)).

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
