# Peer review of "On the estimation of stratospheric age of air from correlations of multiple trace gases"

_EGUsphere, 2024_

## Referee Comment (RC2)

Review of the manuscript

**On the estimation of stratospheric age of air**
**from correlations of multiple trace gases**

submitted to EGUsphere by F. Voet et al.

S. Chabrillat, BIRA-IASB, November 2024

**General Comments**

This study is an exhaustive and potentially very useful contribution on the estimation of stratospheric Age of Air (AoA) from coincident retrievals of several long-lived tracers, using the compact correlations which can be established in model space between AoA and such tracers. It convincingly shows how these new AoA derivations would have significantly reduced uncertainties compared with the standard derivations which rely only on $SF_6$ measurements. It relies on an advanced chemistry-transport model which is well suited to the task, as demonstrated by many earlier papers on AoA and the Brewer-Dobson Circulation (BDC). The methodology is explained in detail, even though its terminology deserves some additional attention. The estimation of uncertainties has been carefully considered, which greatly enhances the interest of this work. The method is applied to both synthetic and actual balloon retrievals, allowing in-depth discussions of the merits of this approach - even though they sometimes feel hastily written and should have been reviewed for consistency.

Overall, I wholeheartedly recommend publication after minor revisions addressing the comments below (especially MC1). I have one additional concern though: the availability of the data. The whole paper is very focused on the creation and usefulness of Look-Up Tables (LUT) between the AoA and tracers, including the associated uncertainties. These tables were derived from CLaMS simulations which have been run from 1979 until 2022, and their applicability is demonstrated from 2011 onwards. Since they seem easy to apply to several existing remote-sensing (and maybe even in-situ) observational datasets, they could be of great interest to the whole community studying the BDC - allowing new comparisons and advancing the whole field. But this cannot be done with the submitted manuscript, because the correlations are shown in the supplement in a graphical format (and only for 4 days of 2011) while they are not published in a numerical format.

I believe that these tables should be publicly downloadable, either in their raw form (mixing ratio bins versus mean and 1-sigma of AoA) or as polynomial fits (coefficients and polynomial expressions). This requires only 12 tables per day, and could be done either as a zip-file supplement or by uploading the dataset to a data repository and documenting the resulting DOI. If for some reason the tables are difficult to derive on other years than 2011 (and August 2021, 2022), that single year would still be useful. If only the 4 specific dates chosen for in-depth study can be made available, that could still allow application to several dozen of ACE-FTS profiles. If none of this can be done, then the final statement on Code and data availability should be corrected, specifying that the CLamS model data **and derived LUT discussed in this paper** may be requested by e-mail to the authors.

**Major comments**

**MC1.** What is exactly the upper limit of the validity of this method, and why? No results are shown above 25 km altitude. I understand that AoA-tracer correlations break down somewhere above this limit, but it would be useful to show where and how exactly. More specifically: from Fig. 5 and A4, I would have expected a a sharp increase in the uncertainty at some level around 25 km in the polar regions of Figure 6 (or maybe Figure 8 or 9?). Does it happen higher up? If yes, it would be good to extend upwards at least that figure in order to highlight this fact. If no, this would indicate that something is missing in the estimation of uncertainties as they should reflect the loss of correlation (i.e. usable information) above some altitude.

On the same topic, I am worried by Figure 10: below 21 km there are large differences between the AoA by CLaMS and by "GLORIA new method". This is good, showing that the new method preserves the information contained in the GLORIA retrievals. But above that limit, the dashed and greenish profiles converge with the solid magenta profile and all three reach a suspiciously good agreement. This is actually mentioned in the discussion (lines 435-437):

> "...the correlations used in the proposed new age calculation method are based on model simulations and not necessarily reflect the actual atmospheric conditions. This could also, to some degree, cause the agreement between the "GLORIA-B new method" AoA and the "CLaMS clock tracer" AoA at the upper end of the height scale."

Could it rather be due to the loss of correlation between AoA and tracers for AoA>4 years (e.g. Fig. 3-4), leading to the "GLORIA new profile" losing any information from the 6 retrievals and simply reflecting instead the model-based correlations for such "old" AoA? That would be unexpected from the in-depth discussions about uncertainties (Fig. 6-9). I believe that to elucidate this point, it would help to plot figures similar to fig. 1-9 but for the two dates of the GLORIA soundings.

**MC2.** The Upper Boundary Condition (UBC) for $SF_6$ plays an important role in this study where it is located at the stratopause (55km). This is explained (lines 145-147) as

> "The upper boundary condition for times outside of the measurement period was created by parameterizing the depicted seasonal cycle of each latitude with a sinusoidal least square fit and adding it to a shifted tropospheric tropical time series (taken from the lower boundary of SF6)."

If that parameterization is as simple as described, I have some concerns. The Semi-Annual Oscillation already plays an important role at the stratopause, and it is modulated by the Quasi Biennal Oscillation (see e.g. Garcia et al., 1994). Or maybe that your sinusoidal least square fit includes several frequencies? Please clarify, and insert a reference if this fit was described in more detail in earlier work.

Could you also be more specific with the UBC for HCFC-22 ? Line 155 states

> "An open upper boundary condition has therefore been defined for HCFC-22."

What does this mean? In my favorite (Eulerian) CTM, anything can happen at the upper boundary if no UBC is specified. Maybe a zero vertical gradient (i.e. null flux) is specified?

Tthese concerns could also be addressed by showing the timeseries of the (modelled and observed) $SF_6$ as well as the (modelled) HCFC-22 at the uppermost level.

**MC3.** What are the limitations of deriving LUT with only one model driven by one reanalysis dataset? Lines 443-444 state

> "*It should be noted that this slow bias of the ERA5 circulation in the comparison presented here is independent of the age calculation method used*".

…and I agree: a biased circulation in the driving reanalysis should impact the long-lived tracers and the clock tracer in the same manner, thus delivering the same correlations as a "perfect" reanalysis. On the other hand, the previous paragraph states (lines 435-436):

> "*...the correlations used in the proposed new age calculation method are based on model simulations and not necessarily reflect the actual atmospheric conditions.*"

…and I also agree with this, but the two sentences seem to contradict each other. Hence I recommend to discuss these "existential questions" a little further in Section 4.

**Specific Comments**

Original text is copied in *italics*, suggestions for corrections are typed in **bold**.

**SC1.** Terminology: some recurring words lead to confusion and should be replaced throughout the text:
- If I understand well, the midpoints of the AoA/vmr bins are interpolated with a polynomial fit which uses the Savitzky-Golay-Filter to create smoother LUT. The result is variously described as "*interpolated series*", "*interpolation*", "*smoothed series*", "*polynomial fit*" or "*Savitzky-Golay-Filter*". Please use a consistent term throughout the text and in the figures ("polynomial fit" is fine) and provide a reference on the Savitzky-Golay-Filter or its usage for this type of procedure (line 206).
- The CLaMS AoA derived from the lag-time of the clock tracer is written very often as *the "true" AoA* or *the "exact" AoA*. Even with the quotes, this is misleading since this AoA is biased by ERA-5… In most cases no adjectives are necessary as it is perfectly clear which AoA you are using as reference. But if an adjective is necessary, I recommend to write systematically "*the **actual** AoA from the clock-tracer*".
- From lines 211 to 308 there are 5 attributions of AoA spread in a mixing ratio bin to "*natural variability*". This is misleading for the same reason: this variability arises from the model simulation and its driving reanalysis and is necessarily natural… Hence I recommend to replace "*natural variability*" by "**model** *variability*" or "**model/reanalysis** *variability*".
- "*one sigma environment(s)*" → "**one-sigma range(s)**"

**SC2.** Abstract, line 6:
> "*this method works well* **in most of the lower stratosphere** *up to a height of about 25 km*"

(in order to account for the white areas in Fig. 6 to 9).

**SC3.** Abstract, line 8-9:
> "*The* **multi-species** *age calculation method is evaluated in a model environment and compared against the*  **actual** *model age* **from an idealized clock tracer**."

**SC4.** The introduction is well written, but please cite some previous papers using compact correlations between AoA and long-lived tracers to derive an estimation of the AoA, and explain how their aims differ from those in your study. To the best of my knowledge, the first such paper was written by **Linz et al. (2017)** and the latest one was written by **Dubé et al. (2024)**. Even though the latter is still a preprint, it is important to cite it because it also uses CLaMS.

**SC5.** Section 2.1: please provide some details about modelling of vertical circulation in CLaMS. I guess that these are radiative heating rates between the isentropes, but do they come from ERA-5 or from an internal part of CLaMS? This is important as in each case there could be biases in these heating rates (which could be mentioned later in the discussion of the results).

**SC6.** Legend of Figure 2: the details of this Figure are already explained in its lower text box and in the text, so I believe that it is not necessary to repeat them for a third time in this Legend (also because all terminology issues arise there - see SC1). A shorter legend could be:

> "***Figure 2.*** *Schematic representation of method used to estimate the* **AoA corresponding to measured mixing ratios and its associated** *uncertainty* **(see text for details)**."

**SC7.** Line 219: " *...added together through means of (Gaussian) error propagation...*"
→ do you mean that they are added as a Root Mean Square Error (RMSE) ?

**SC8.** Line 268, first sentence introducing Fig 3: please clarify here that you are not yet showing the binning hemispheric approach described in the earlier section 2.2.

**SC9.** Line 270:
"*Additionally, the equivalent latitudes of the model points* **(Nash et al., 1996)** *are color-coded…*"

**SC10.** Lines 312-317: this procedure is written in a rather obscure manner, even though it seems quite simple. Please re-formulate, e.g. explaining that prior to the application of these LUT you want to compare the 6 total uncertainties for each species among each other and also with the total combined uncertainty.

**SC11.** Line 337: "*(compare* **with** *Fig. A4)*".
This figure A4 is compared multiple times with Fig. 5, 6, 7. I think that it should be moved from the Appendix to the main text.

**SC12.** Lines 355-356:

> "*The pseudo-measurements were created by adding normally distributed random noise to the mixing ratios of the six trace gases for all air parcels on the four considered days.*"

These synthetic measurements are not completely realistic as they do not take systematic errors into account. This is a real concern and should be mentioned somehow, because actual measurements would be biased w.r.t. modelled mixing rations since there are biases in the ERA-5 circulation.

**SC13.** Line 362: " *The thick black lines represent the zonally averaged tropopause from the ERA5 reanalysis data for each day*". Is this the thermal (T gradient) or dynamical (PV+theta) tropopause?

**SC14.** Lines 365-367: "*Even the northward intrusion of air with AoA below one and a half years into the layer of air with AoA between one and a half and two years at roughly 70°N and 14 km height in July is clearly present in the results of the correlation method (compare Figs. A4 (c) and 7 (c))*"
This region is not obvious to identify at first. Consider helping the reader by drawing a bounding box around it, at least for these two figures.

**SC15.** Lines 385-386: you mention Figure 8 but shouldn't that be Fig. 9 there ?

**SC16.** Lines 385-388:

"*(Note that the scales of the colorbars are different and that the contours show different things in the two figures)*"

This is the straw that broke the camel's back. I was wondering since Fig. 8 about the added value in showing absolute differences with color coding and relative differences with contour labels, which is quite confusing and was not done for fig. 6. Note that fig. 6 also could have shown absolute errors in color shading and relative errors with contour labels, but I am not complaining that this was not done. On the contrary, I really think that the color scales and/or contour labels should be changed in Fig. 8 and 9 to simplify them and allow direct comparison with fig. 6 !

**SC17.** Lines 395-401: please synthesize in a few sentences

"*...the standard convolution method, as described in Garny et al. [under review], and the subsequent correction for SF6-depleted air from the mesosphere introduced by Garny et al. (2024).*"
…including the limitations of this method and this correction. This is important because these limitations play a role in the discussion of Fig. 10 while the review by Garny et al. is still being reviewed.

**SC18.** Figure 10: the dashed "*greenish*" lines and the greenish shadings are not really visible, especially on top of the blue shadings. Consider using black lines and black horizontal error bars instead (possibly for a subset of the levels in the case of horizontal error bars).

**SC19.** Lines 430-432: " *Another possibly explanation  **would** be  some issue with the instrument during the Timmins flight lead**ing** to the retrieval of systematically too low SF6 mixing ratios. Perhaps the unintended descent of the gondola down to 22 km mentioned in sect. 2.3 could have caused such an issue.*"
→ … while not damaging the retrievals of the 5 other species, leading to the very different values by "GLORIA new method" ??

**SC20.** Line 462: "*The lookup tables for the remaining cases can be found in Figs. S17 to S28 of the supplement to this article*" → these are not LUT, they are plots of the LUT (see MC1).

**SC21.** Figure 11(f): "*...by means of gaussian error propagation). Contours: Relative difference between color-coded values and AoA from clock tracer.*"
This is very unclear… What do these contour lines mean exactly? Are they necessary? They are not discussed in the text…

**SC22.** Section 4.2: this discussion on the stability of the correlations is quite confusing:

"*Since these depletion processes do not fundamentally change over time, the future correlations of the five mentioned trace gases with AoA will likely be similar to the way they were in 2011, the year considered in this study*".

But GLORIA-B used LUT made specifically for its flight dates, so this question about the stability of the correlations is moot: the model correlations can be computed for the days of the measurements, i.e. in much more recent years tha 2011, and this was actually done! It looks to me like this section was written before the comparison with GLORIA-B and not removed afterwards.

**SC23.** Line 574: "*Also, such datasets could be used to study exchange processes between the troposphere and the stratosphere and therefore help to better constrain new emissions of prohibited substances like CFC's*".

This sentence is not clear. There are many other good reasons to sudy troposphere-stratosphere exchanges.

**Typos, wording etc.**

- Line 18: "*... Brewer-Dobson circulation (***BDC; see e.g.** *Holton et al., 1995; Butchart, 2014*)
- Line 78: "*... if the***y** *can be retrieved...*"

- Lines 198-205:
  " **For visual clarity***, only the last three of these mixing ratio bins are shown in the figure. The histogram illustrates the distribution of AoA within a given mixing ratio bin. The blue area in the histogram highlights the one sigma environment around the mean AoA value of the distribution (mean value ± one * **standard deviation***). Similarly, the blue area in each of the three depicted mixing ratio bins corresponds to the one-sigma * **range** *around the mean value of the respective AoA distribution inside. Such one-sigma * **ranges** *are calculated for each one of the one hundred fifty mixing ratio bins. Subsequently, a midpoint for each bin with the sample mean AoA as the x- and the middle of the bin range as the y-coordinate was then defined. Th***ise** *constructed set of midpoints constitutes a sort of look-up table that can be used to interpolate a***n** *AoA value...*"

- Line 216: first sentence actually belongs to previous paragraph.

- Lines 372: "*The absolute difference  reach***es** *its highest values...*"

- Lines 381-382: "*These standard deviations are a quantification of the spread of the AoA difference in zonal direction. They can be used to estimate the uncertainty of AoA derived from individual measurements at different * **longitudes** *for any * **latitude.**"

- Line 401: first sentence actually belongs to previous paragraph.
- Line 410: "*The magenta shading around the "GLORIA-B new method " AoA represents…*"
- Line 425: "*…however, any conclusions drawn from the  **corresponding** values are hardly meaningful*"
- Lines 451-472: please replace all occurrences of "3d" with "3-D".

**Additional bibliographical references**

Garcia, R. R., T. J. Dunkerton, R. S. Lieberman, and R. A. Vincent (1997), Climatology of the semiannual oscillation of the tropical middle atmosphere, *J. Geophys. Res.*, 102(D22), 26019–26032, doi:10.1029/97JD00207.

Nash, E. R., P. A. Newman, J. E. Rosenfield, and M. R. Schoeberl (1996), An objective determination of the polar vortex using Ertel's potential vorticity, *J. Geophys. Res.*, 101(D5), 9471–9478, doi:10.1029/96JD00066.

Linz, M., Plumb, R. A., Gerber, E. P., Haenel, F. J., Stiller, G., Kinnison, D. E., Ming, A., and Neu, J. L.: The strength of the meridional overturning circulation of the stratosphere, Nature Geoscience, 10, 663–667, https://doi.org/https://doi.org/10.1038/ngeo3013, 2017.

Dube, K., Tegtmeier, S., Ploeger, F., and Walker, K. A.: Hemispheric asymmetry in recent stratospheric age of air changes, EGUsphere [preprint], https://doi.org/10.5194/egusphere-2024-1736, 2024.

---

## Community Comment (CC1)

Review for the EGU Peer Review Training

"On the estimation of stratospheric age of air from correlation of multiple trace gases"

F Voet., et al

**Note**: I compiled this review in the framework of the Copernicus ECS peer-review training 2024.

**Overall Impression**

This research introduces a novel method to estimate the Age of Air (AoA) by analyzing atmospheric measurements of various long-lasting trace gases. The method is based on examining the relationships between specific trace gas species (such as CFC-11, CFC-12, HCFC-22, CH4, N2O, and SF6) and AoA. The effectiveness of this method was tested using simulations with the CLaMS model and resulted in a calculated weighted mean AoA from satellite observations of the six trace gases. The study found that the difference between this calculated AoA and the actual AoA in the model remained below half a year in the lower stratosphere. Additionally, the method was applied to measurements using GLORIA-B, leading to more reliable results and significantly reduced uncertainty compared to traditional methods. Overall, this study demonstrates the potential accuracy of determining AoA from satellite observations of multiple trace gases and their correlations with AoA.

The authors have successfully highlighted the need and relevance of the study. They have very well explained the setup of the model, the calculations done for AoA estimation and data analysis from GLORIA-B instrument in the methods section making their study reproducible. Results are clear and appear in a logical manner with all the figures being explained in detail with comprehensive captions. The authors have also suggested future improvements for their new method.

Based on my review of the manuscript and considering the comments below, I suggest minor revisions. The manuscript is overall very well written and structured. I would like to inform the editors that the study topic is new to me, as my expertise lies in atmospheric chemistry in the troposphere. Specifically, I work with the molecular-level chemical composition of sea spray aerosol and the associated aging processes through laboratory experiments and field campaigns. I have extensive experience using a chemical ionization mass spectrometer and can provide better reviews for papers in the field of atmospheric chemistry.

However, I have still made my best efforts to review the given manuscript but there could be some potential errors that I could not recognize.

**Minor Comments**

Abstract – The method for mean age calculation should be described more clearly. I am not able to understand exactly the meaning of compact correlations. Do they mean strong positive correlations between the trace gases and AoA? Also, the authors should consider giving a standard name to their method.

Figure 2: expand vmr (volume mixing ratio) in the figure caption to improve readability

Tables 1 and 2 could be combined for conciseness.

Line 48: Spelling error - CO2 **possess** a strong seasonal cycle in tropospheric mixing ratios……

Line 78: Spelling error - if **they** can be retrieved with sufficiently low uncertainty (Schoeberl et al., 2005).

Lines 107-110: I suggest to modify these sentences for consistency. I have highlighted the part of the sentence that could be changed. For example,

From - "The lowest model layer (lower boundary layer of the model) extends from the surface to approximately 1.5 km (more precisely0 K $< \zeta <$ 70 K). The uppermost model layer (upper boundary layer) covers the potential temperature range $2350 - 2650$ K (altitude of about 55 km), hence the model domain extends from the surface to about the stratopause."

To - "The lowest model layer (lower boundary layer) extends from the surface to approximately 1.5 km (more precisely 0 K $< \zeta <$ 70 K). The uppermost model layer (upper boundary layer) **extends up to an altitude of about 55 km (potential temperature range $2350 - 2650$ K)**, hence the model domain extends from the surface to about the stratopause."

Line 158: Stylistic error with the use of inverted commas - "exact" should be "exact". This error is observed at several instances throughout the manuscript, such as, lines 186, 195, 401, 415, 421, and Fig 10 caption.

Line 263: The authors could briefly describe the convolution method since the reference is still under review. This would help the readers to understand your method better.

Figure 10: It is a very intriguing plot which has been well explained in the text. I understand that the new correlation method has improved the Gloria standard method. I also understand that the CLaMS new method is comparable with the CLaMS clock tracer method, making the correlation method reliable. However, I am wondering why the use of the new method is increasing the distance between the Gloria measurements (magenta line) and CLaMS model results (dashed magenta) compared to the standard Gloria method (blue line). What could be a potential explanation of observations and model results becoming more distant between 18-21 km when using the new method?

---

## Community Comment (CC2)

**On the estimation of stratospheric age of air from correlations of multiple trace gases**

By F. Voet et al.

**Note:**

As an early career researcher, I participated in the Copernicus peer-review training 2024. This community comment, presented in the form of a referee report, is the outcome of that training. I would like to disclose that I have a potential conflict of interest, as I am co-authoring scientific papers with several of the authors of this manuscript.

**General comments**

This work introduces a new method to derive mean age from remote-sensing observations. This method uses model-derived correlations between six individual trace gases and mean age to create lookup tables, enabling the inference of a weighted mean age from remote-sensing data. The authors show that by using simultaneous measurements of multiple trace gases, the uncertainty of the derived mean age values can be reduced significantly compared to the conventional method that uses one trace gas. The method is applied to data from a balloon flight, and the results are compared to model outputs and to those obtained using the conventional method.

This study provides a valuable and timely contribution to the field of stratospheric transport times. The authors convincingly argue that at least one upcoming satellite mission will likely provide data to which the new method could be applied to in future research, thereby unleashing the full spatio-temporal potential of satellite data for mean age estimation.

The authors provide a thorough uncertainty estimation that emphasizes the value of their new method.

I find the manuscript to be very well-structured and comprehensively written.

On the basis of my read and in light of the comments below, I recommend that the article be accepted, subject to minor revisions.

**Main comments**

MC1: The abstract is the most-read part of the manuscript. In light of this, I suggest improving the flow to better guide the reader through your study. For example, the abstract mentions satellite measurements very early, even though the new method is not applied to satellite data. I understand that the anticipated availability of satellite data in the near future is the main motivation for implementing this new method. However, I suggest either placing the satellite part later in the outlook section of the abstract or keeping it early but explicitly mentioning that the method has not yet been applied to satellite data. Also please see SC1 for another suggestion.

MC2: While I appreciate that you have included the correlations and lookup tables in the supplementary material for completeness, I believe that it is too much information for the supplementary document. Please consider uploading these images and the corresponding data to a separate repository with a unique DOI.

**Specific comments**

SC1: Abstract lines 6-7: To describe the background of the method in a more precise way, you could mention, that die correlations are derived from the model, e.g. "The method is based on the compact correlations of these gases with mean age in the *model world*."

SC2: l 113-114: For me as a non-modeler, this sentence is difficult to follow: "The values of these time series are interpolated latitude-wise onto the Lagrangian air parcels inside the lower boundary layer at the beginning of each new simulation time step." Do you mean "At the beginning of each simulation time step, the values from these time series are interpolated across latitude onto the Lagrangian air parcels within the lower boundary layer."?

SC3: l 147-149: Did you shift the lower boundary of $SF_6$ by a constant, i.e. did you use a long-term average of the mean age values at 2500 K derived from MIPAS measurements? Or did you use a varying time shift? Please clarify.

SC4: l 263: I suggest clarifying "the $SF_6$ sink correction method described in Garny et al (2024) …". This may avoid confusion if the reader stumbles across the similar terms "correction method" and "correlation method".

SC5: Figure 3: I suggest aligning the y-axis scale to that the individual panels are easier to compare.

SC6: Figure 6 caption: This is an extremely useful figure, congratulations! The caption could be improved however. AoA from clock tracer below one year is attributed as tropospheric air, which is not accurate, since there is obviously stratospheric air involved. too. Do you mean "highly influenced by tropospheric air"? The same applies to Figure 7, 8 and 9 captions

SC7: Throughout the manuscript: I suggest giving the "new method" and the "standard method" a name, so that it is easier to reference in future research. You could ask AI to create an acronym for you. My search suggested e.g. "STAGE – Single-Tracer AGe Estimation" and "MCAGE – Multi-Tracer Correlation-based AGe Estimation".

SC8: Throughout the manuscript: please double check for consistent use of "AoA", "mean AoA" and "mean age". AoA == Mean age?

SC9: l 491 onward: I suppose by "functional relations" you mean the correlations, that your method is based on? I suggest to be consistent here for clarity.

SC10: l. 596 onward: This paragraph could be improved for clarity using a more concise language in order to better guide the reader. As a start, in line 501-502, you could consider adding the new method's name (e.g. MCAGE), as this method is what requires the lookup table that this paragraph is about.

SC11: Figure 11 discussion, l. 458 onward: How did you generate your zonal mean? Did you use equivalent latitude? Or are you averaging over longer time periods, so that equivalent latitude will be the same as latitude?

SC12: l. 510: By "zonal mean trends" do you mean "zonal mean trace gas concentrations"? Please clarify.

---

## Author Comment (AC1)

**Review of the Manuscript**

**Overall Comments**

Overall I think that this is an interesting and well-written paper which is very suited to ACP. The paper describes a new method for estimating age-of-air which offers reduced uncertainty compared to using a single trace gas such as SF6. The method is described and illustrated in some detail. I think the paper will be suitable for publication after addressing the points below. My main comment is that paper uses modeled tracers to illustrate the method and I am not (at present) convinced that all sources of uncertainty have been accounted for. I.e. what about possible systematic errors in the model's ability to reproduce one or more of the individual trace gases used due to errors in kinetic parameters.

We sincerely thank the reviewer for their interest in our manuscript, their thorough evaluation, positive feedback, and constructive criticism. We have addressed all the major and minor comments in the revised version of the manuscript and provide a point-by-point reply below (with reviewer comments in black and replies in blue). The main changes in the revised manuscript are as follows:

(a) We slightly improved the AoA calculation method further by including an additional condition for the construction of the lookup tables to account for the deterioration of the tight correlations at lower mixing ratios,

(b) We added a comparison of the AoA-profiles derived from the GLORIA-B measurements with recently published AoA-profiles of an independent study (`https://egusphere.copernicus.org/preprints/2024/egusphere-2024-3279/`)

(c) We added a more thorough discussion of model uncertainties in the new section 4.4 "Model and method uncertainties", where uncertainties and possible biases of the CLaMS model are addressed (included in this document as section 4.4).

**Main Points**

1) A major aim of the paper is to argue that the new method will reduce uncertainty. However, quantification of the errors (e.g. uncertainty below 0.3 yrs as given in the abstract) is based on a model where it is possible to get the best possible agreement between the AoA from different methods. I think that the quoted uncertainty is mainly based on atmospheric variability. What about uncertainty and bias in the modeled tracers due to e.g. photochemical data, model-calculated loss rates. Related to that, how well does the model reproduce the individual tracer profiles (and tracer- tracer correlations) from the observations such as GLORIA. Please discuss these points.

We agree that we need to address possible uncertainties and biases of the CLaMS model in more detail. We have therefore included a comparison of the AoA-profiles we have derived from the GLORIA-B measurements with the recently reported profiles of an independent

study. The comparison was added as follows to Sec: 3.4 "Application to GLORIA balloon data":

> *The two profiles of "balloon borne Gimballed Limb Observer for Radiance Imaging of the Atmosphere (GLORIA-B) new method" are also in good agreement with the recently published mean age of air (AoA) profiles by Schuck et al. [2024b]. In their study, Schuck et al. [2024b] calculated AoA values using cryogenic whole-air samples of sulfur hexafluoride ($SF_6$) and carbon dioxide ($CO_2$) collected in Kiruna, Sweden, during August 2021. Similar to the AoA profiles of "GLORIA-B new method" the results reported by Schuck et al. [2024b] exhibit an increase in AoA with altitude, reaching approximately five years at around 22 km. Beyond this altitude, the AoA values remain approximately constant at their maximum of five years. There are good reasons to believe that the utilized model-based lookup tables do reflect the actual atmospheric conditions well enough to justify their application to the GLORIA-B measurements. These reasons will be discussed in the next section. On top of that, an alternative to the model in the form of global satellite measurements as the foundation of the lookup tables will also be discussed in the next section. (Lines 475 to 483 in revised manuscript)*

Additionally, we have introduced a new discussion section, "4.4 Model and Method Uncertainties," where we address the applicability of the improved model-based lookup tables to the measurements of GLORIA-B (added as Sect. 4.4 in this document). In this section, we highlight that previous studies have demonstrated the reliability of the Chemical Lagrangian Model of the Stratosphere (CLaMS) model for the trace gases used. We also emphasize that while errors in the model's kinetic parameters, which could introduce biases in the results, cannot be fully ruled out, a comprehensive analysis to quantify these uncertainties would extend the scope of our study.

2) The model simulation used here applies a HALOE-based mid-stratosphere upper boundary condition for the (non-zero) tracers. This will:

   (a) Impose a circulation speed on the tracers that may be inconsistent with the model dynamics (e.g., as pointed out for ERA5), and

   (b) Limit the use of this method to a period for which such boundary data could be derived.

   Please discuss the impact of these points.

   In general, we agree with the referee about the way the HALOE-based upper boundary condition could cause inconsistencies with the model dynamics as described in (a). However, the impact of this upper boundary condition on our results is expected to be very small, as effects of changes in the upper boundary on considered trace gas composition below 25 km are almost negligible. We evaluate the influence of the upper boundary of the model in the newly added section "4.4 Model and method uncertainties" in the discussion of the revised manuscript (added as Sect. 4.4 in this document) and added a short discussion on the limitations of our study there.

**Minor Points**

1) Figure 1 caption and elsewhere. "greenish". This is not a colour. Please choose an actual colour and use that for lines and descriptions.

The color of the profile "CLaMS clock tracer" and its uncertainty range in Figs. 1 and 10 (original manuscript) was changed to black in the revised manuscript.

2) Page 5, Line 109: The range 0 K to 70 K. Please explain more clearly the parameter you are using to determine the boundary layer up to 1.5 km.

We have added more information about the hybrid vertical coordinate $\zeta$ and clarified that $\sim 1.5\,\text{km}$ was chosen as the height of lower boundary of the model, and it roughly corresponds to the height of the planetary boundary layer

> $\zeta$ is equal to the potential temperature $\theta$ above a predefined level, and gradually turns to zero at the surface. Independent of the elevation of the surface, $\zeta$ is defined in a way that it is equal to zero at every surface point [for details see Pommrich et al., 2014]. The lowest model layer, representing the lower boundary layer of the model, extends from the surface to approximately 1.5 km (specifically, $0\,\text{K} < \zeta < 70\,\text{K}$), which roughly corresponds to the height of the planetary boundary layer. (Lines 114 to 118 in revised manuscript)

3) Page 6: These loss reactions will, to varying degrees, depend on temperature (including T-dependent cross-sections) and the overhead column ozone. How is the OH field calculated? What about loss rates in the troposphere (e.g., tropospheric OH)? Please explain if and how these variations are taken into account, and the implications if they are not (e.g., increased uncertainty if these factors affect different tracers in different ways).

The mechanism is in principle explained in the paper by Pommrich et al. (2014). Photolysis reactions are calculated as diurnal averages by the DISSOC photolysis code using overhead ozone from a climatology, in this case derived from HALOE observations (Grooss and Russell, 2005). OH, O(1D), Cl and Br are taken from a climatology created with the Mainz photochemical 2D model [Grooß, 1996] as diurnal average mixing ratios. This climatology had been pre-calculated for 18 latitudes, 34 pressure levels and 12 months by the CLaMS code. To clarify this we revised manuscript as follows:

> For the use in the present study, the additional trace gas species sulfur hexafluoride ($SF_6$) and HCFC-22 have been implemented into the CLaMS model similar to the method described by Pommrich et al. [2014]. The depletion mechanisms for HCFC-22 implemented in CLaMS is represented by the reactions ... (Lines 131 to 133 in revised manuscript)

4) Pages 6 and 7: Please clarify the length of the model simulation. It must extend to 2022 for the GLORIA flights but, for example, HALOE cannot be providing the upper boundary conditions through to this date.

The way in which the upper boundary conditions for $N_2O$ and $CH_4$ were created over the entire simulation period is now specified in section 2.1 "Setup of the Chemical Lagrangian Model of the Stratosphere CLaMS":

> Similar to Pommrich et al. [2014], the climatology of the Halogen Occultation Experiment [Grooß and Russell, 2005] was used as the upper boundary for methane ($CH_4$) in this study. More specifically, the mean seasonal cycle from the Halogen Occultation Experiment climatology was used for every year of the simulation period. The Mainz photochemical 2D model [Grooß, 1996] was used for the upper boundary of nitrous oxide ($N_2O$). (Lines 159 to 162 in revised manuscript)

5) Page 10, Table 2: It would be good to include some estimates of the stratospheric lifetimes of $CH_4$ (much longer than overall total) and $N_2O$ (very similar to overall total) from other sources.

*The stratospheric lifetimes of $N_2O$ and $CH_4$ are now added to table 2.*

6) Page 16, Line 347: It is confusing to say "boreal winter season" and then point to results at high southern latitudes (which is in the summer). Please rephrase.

*The passage has been rephrased as follows:*

> *On average the uncertainties appear highest for austral summer season (1 January) at high southern latitudes, which is likely due to the collapse of the Antarctic polar vortex a few months prior, and the subsequent gradual spread of air masses that were contained in the vortex before its collapse.* (Lines 379 to 382 in revised manuscript)

7) Page 17, Figure 7 caption: Explain what definition is being used for the tropopause (T, lapse rate, PV, etc.).

*The following clarification has been added to the caption of Fig. 7 (original manuscript) and the captions of all other figures depicting the tropopause:*

> *(World Meteorological Organization (WMO) definition by lapse rate from 1957)*

**Typos**

1) Page 2, Line 48: "posses" → "possesses".

2) Page 4, Line 78: "the" → "they".

3) Page 6, Line 123: Use a small "s" for "sulfur".

4) Page 10, Line 232: Add a space after the closing parenthesis.

5) Page 13, Line 289: The lack of meridional gradients will be partly due to the much-decreased emissions by 2011. This should be pointed out.

6) Page 12, Line 288: "Lifetime" (remove the space).

7) Page 13, Figure 4 caption: "25 km in southern hemisphere" (not "on"). Use a small "p" in "part".

8) Page 13, Line 293: Add "section" for section 2.2.

9) Page 14, Line 314: Change "pretending" to something like "assuming".

10) Page 15, Line 340: Change "high" to "large".

11) Page 22, Line 432: "descend" → "descent".

*Thank you for carefully watching out for typos in the manuscript. They have all been corrected for in the revised version.*

**4 Discussion**

**4.4 Model and method uncertainties**

This study should, first and foremost, be viewed as a proof of concept for the proposed method to calculate AoA. Such a proof of concept can only be demonstrated within the perfectly self-consistent environment of a model, where the "true" values of the desired quantity are already known. Since no model is a perfect representation of the actual atmosphere, the lookup tables required for applying the proposed method should ideally be constructed directly from measurements, as laid out in Sect. 4.1, rather than from model results. Due to the lack of suitable measurement data, we had to establish the lookup tables required for applying the method to the GLORIA-B data using the available model results. We are confident that the CLaMS-based lookup tables used on the GLORIA-B data represent the actual atmospheric conditions reasonably well and find the AoA profiles derived from these lookup tables to be more plausible than the corresponding ones derived with the standard method (see Fig. 11). Our confidence in the CLaMS-based lookup tables stems from the following three reasons:

1. The reliability of the CLaMS model for the utilized trace gases has been demonstrated in previous studies (see, e.g., Laube et al. [2020] for trichlorofluoromethane (CFC-11), dichlorodifluoromethane (CFC-12), and HCFC-22, Konopka et al. [2004] for $CH_4$, and Pommrich et al. [2014] for $N_2O$).

2. There is excellent agreement between the AoA profiles reported by Schuck et al. [2024b] and those derived using the proposed method from the GLORIA-B measurements at Kiruna (see Fig. 11). The balloon-borne air samples collected by Schuck et al. [2024b] should closely resemble the air masses observed by GLORIA-B during its first flight, as both balloon flights took place in August 2021 in Kiruna, Sweden. The fact that Schuck et al. [2024b] obtained similar results for comparable air masses using a different calculation method strongly supports our expectation that the utilized CLaMS-based lookup tables reflect the conditions of the actual atmosphere reasonably well.

3. The upper boundary region, arguably the biggest source of bias in the simulation as a whole, seems to have only limited impact on the model tracer mixing ratios in the considered altitude region (i.e., below 25 km). Figure A4 in the appendix shows the temporally averaged zonal mean $SF_6$ distribution of the model results up to the upper boundary over two different three-month periods. Fig. 11a shows the average distribution from December 2010 to February 2011, representing winter in the Northern Hemisphere, and Fig. 11b shows the average distribution from June 2011 to August 2011, representing winter in the Southern Hemisphere. The winter periods were chosen because they represent the time when the downward transport of air from higher altitudes by the deep branch of the Brewer–Dobson circulation (BDC) is strongest in the respective hemisphere. $SF_6$ was selected because, in relative terms, it reaches the upper boundary in higher amounts than all other tracers (due to the absence of a stratospheric sink). Out of all model results, the influence of the upper boundary is therefore expected to be strongest for $SF_6$ during winter in the respective hemisphere. Figure 11 suggests that the downward transport 11, the downward transport of $SF_6$ depleted air from the upper boundary only seems to have an affect below 25 km at high latitudes in the winter hemisphere. This limited influence of the upper boundary on model results below 25 km could partly be related to the slow bias of the stratospheric European Centre for Medium-Range Weather Forecasts Reanalysis v5 (ERA5) circulation, as pointed out in sect. 3.4. This slow bias, however, effects the six long-lived trace gases in the same way as the model clock tracer and therefore can't be responsible for any biases the model-based lookup tables might have.

While these considerations give us confidence in the plausibility of the CLaMS-based lookup tables, it is important to emphasize that a thorough analysis would be necessary to fully assess the influence of the upper boundary on each of the six trace gases specifically, and to evaluate the extent to which the derived lookup tables reflect the conditions of the real atmosphere. Such an analysis would ideally involve the use of several atmospheric models and/or reanalysis datasets to better quantify potential biases and uncertainties. Additionally, the influence of uncertainties in the kinetic model parameters, such as photochemical data and model-calculated loss rates, on AoA cannot be ruled out. A thorough analysis of the influence of these uncertainties on AoA would also be needed. However, conducting such extensive analyses is beyond the scope of this study.

---

## Author Comment (AC2)

**Review of the Manuscript**

**On the Estimation of Stratospheric Age of Air from Correlations of Multiple Trace Gases**
submitted to EGUsphere by F. Voet et al.

**Reviewer: S. Chabrillat, BIRA-IASB, November 2024**

**General Comments**

This study is an exhaustive and potentially very useful contribution on the estimation of stratospheric Age of Air (AoA) from coincident retrievals of several long-lived tracers, using the compact correlations which can be established in model space between AoA and such tracers. It convincingly shows how these new AoA derivations would have significantly reduced uncertainties compared with the standard derivations which rely only on SF6 measurements. It relies on an advanced chemistry-transport model which is well suited to the task, as demonstrated by many earlier papers on AoA and the Brewer-Dobson Circulation (BDC). The methodology is explained in detail, even though its terminology deserves some additional attention. The estimation of uncertainties has been carefully considered, which greatly enhances the interest of this work. The method is applied to both synthetic and actual balloon retrievals, allowing in-depth discussions of the merits of this approach - even though they sometimes feel hastily written and should have been reviewed for consistency.

Overall, I wholeheartedly recommend publication after minor revisions addressing the comments below (especially MC1). I have one additional concern though: the availability of the data. The whole paper is very focused on the creation and usefulness of Look-Up Tables (LUT) between the AoA and tracers, including the associated uncertainties. These tables were derived from CLaMS simulations which have been run from 1979 until 2022, and their applicability is demonstrated from 2011 onwards. Since they seem easy to apply to several existing remote-sensing (and maybe even in-situ) observational datasets, they could be of great interest to the whole community studying the BDC - allowing new comparisons and advancing the whole field. But this cannot be done with the submitted manuscript, because the correlations are shown in the supplement in a graphical format (and only for 4 days of 2011) while they are not published in a numerical format.

I believe that these tables should be publicly downloadable, either in their raw form (mixing ratio bins versus mean and 1-sigma of AoA) or as polynomial fits (coefficients and polynomial expressions). This requires only 12 tables per day, and could be done either as a zip-file supplement or by uploading the dataset to a data repository and documenting the resulting DOI. If for some reason the tables are difficult to derive on other years than 2011 (and August 2021, 2022), that single year would still be useful. If only the 4 specific dates chosen for in-depth study can be made available, that could still allow application to several dozens of ACE-FTS profiles. If none of this can be done, then the final statement on Code and data availability should be corrected, specifying that the CLamS model data and derived LUT discussed in this paper may be requested by e-mail to the authors.

Thank you for your thorough review and valuable feedback on our manuscript. We appreciate your positive assessment and recommendations. We have carefully addressed your comments and made several important revisions to the manuscript. I would like to highlight three main changes:

(a) We have implemented an additional condition for the construction of the lookup tables to account for the deterioration of tight correlations at lower mixing ratios.

(b) We have included a comparison of the AoA-profiles derived from the GLORIA-B measurements with recently published AoA-profiles from an independent study (`https://egusphere.copernicus.org/preprints/2024/egusphere-2024-3279/`). This comparison provides additional validation for our method and places our results in the context of current research in the field.

(c) We have added a new section, "4.4 Accuracy of the CLaMS Model" to the discussion (added to this document under same number and name). This section addresses the uncertainties and possible biases of the CLaMS model, providing a more comprehensive assessment of our methodology's limitations and strengths.

Regarding your concern about data availability, we have taken steps to make the Look-Up Tables (LUT) more accessible. The complete lookup tables, including measurement uncertainty and model variability for the six trace gases, are now available online at: `https://doi.org/10.5281/zenodo.14543944`. We believe these changes significantly enhance the manuscript and address your concerns. Thank you again for your insightful comments, which have helped improve the quality and impact of our work.

**Major Comments**

**MC1**

What is exactly the upper limit of the validity of this method, and why? No results are shown above 25 km altitude. I understand that AoA-tracer correlations break down somewhere above this limit, but it would be useful to show where and how exactly.

We agree with the reviewer that the reason for choosing an upper limit of 25 km for the proposed method is not clearly stated or demonstrated in in the manuscript. We have therefore included CLaMS results between 25 km and 30 km as blue dots in Figs. 4 and A2 and added the following passage for clarification to the description of Fig.4 in section 3.2 "Lookup tables to get AoA from mixing ratio"

> *The blue dots are the same as the green ones, but in the altitude range between* 25 km *and* 30 km*. The comparison between the green and blue dots illustrates the reason for the general altitude limit of* 25 km *for all days and trace gases. A general height limit was chosen only for the sake of simplicity, and* 25 km *as the general height limit seems to assure lookup tables of a reasonable length in all cases. In the case of dichlorodifluoromethane (CFC-12) in the southern hemisphere, for instance, the correlation between tracer mixing ratio and mean age of air (AoA) probably wouldn't have been compact enough to create a lookup table for October beyond AoA around 4 years, if data between* 25 km *and* 30 km *had been included (see Fig. 4 (d)).*

(Lines 335 to 341 in revised manuscript)

More specifically: from Fig. 5 and A4, I would have expected a a sharp increase in the uncertainty at some level around 25 km in the polar regions of Figure 6 (or maybe Figure 8 or 9?). Does it happen higher up? If yes, it would be good to extend upwards at least that figure in

order to highlight this fact. If no, this would indicate that something is missing in the estimation of uncertainties as they should reflect the loss of correlation (i.e. usable information) above some altitude.

As the reviewer correctly points out, a sudden increase in the uncertainty values plotted in Fig. 6 does indeed occur above 25 km in the polar regions. This increase is caused by the fact that the weighted mean uncertainty abruptly reflects only the high uncertainty of $SF_6$, as the other tracers correlate less tightly with AoA and therefore cannot be used to the same extent for its calculation. However, the AoA range of the lookup tables for $SF_6$ is only marginally longer than those for the other trace gases. For this reason, extending the plots in Fig. 6 above 25 would rather reveal areas of missing values than areas of high uncertainty. This fact is illustrated in Fig. 1, an extension of Fig. 6 (b) in the submitted manuscript up to 30 km.

[Figure]

Figure 1: Extension of Fig. 6 (b) in submitted manuscript up to 30 km

More importantly, however, the lookup tables used to estimate the weighted mean uncertainty in Fig.6 are constructed only from CLaMS data points below 25 km. It would therefore be inadmissible to use these lookup tables above 25 km in the first place. To avoid any misconceptions about the intended use of the lookup tables, we have decided to keep the plots in Fig. 6 restricted to 25 km as they were in the original manuscript.

On the same topic, I am worried by Figure 10: below 21 km there are large differences between the AoA by CLaMS and by "GLORIA new method". This is good, showing that the new method preserves the information contained in the GLORIA retrievals. But above that limit, the dashed and greenish profiles converge with the solid magenta profile and all three reach a suspiciously good agreement. This is actually mentioned in the discussion (lines 435-437):

> "...the correlations used in the proposed new age calculation method are based on model simulations and not necessarily reflect the actual atmospheric conditions. This could also, to some degree, cause the agreement between the "GLORIA-B new method" AoA and the "CLaMS clock tracer" AoA at the upper end of the height scale."

Could it rather be due to the loss of correlation between AoA and tracers for AoA>4 years (e.g. Fig. 3-4), leading to the "GLORIA new profile" losing any information from the 6 retrievals and simply reflecting instead the model-based correlations for such "old" AoA? That would be unexpected from the in-depth discussions about uncertainties (Fig. 6-9). I believe that to elucidate this point, it would help to plot figures similar to fig. 1-9 but for the two dates of the GLORIA soundings.

We want to thank the reviewer for bringing a point of concern regarding the original lookup tables to our attention. Some of the original lookup tables do indeed lose information about

AoA for decreasing mixing ratios at certain points. This loss of information is caused by the steepening of the gradient with decreasing mixing ratio, which, at some point, leads to the breakdown of the correlation between mixing ratio and AoA. For mixing ratios below that point, the affected lookup tables show hardly any variation in AoA and essentially only return the maximum AoA value of the CLaMS model (see, e.g., Fig. 4(a) in the original manuscript). To avoid using the lookup tables for mixing ratio ranges where they no longer carry information about AoA, we have devised a condition for truncating the lookup tables and have updated them accordingly. The newly added passage in Section 2.2, "Age of Air Calculation," of the revised manuscript reads as follows:

> *Depending on the trace gas and the day of the year, the correlation between tracer mixing ratio and AoA can eventually break down as mixing ratios decrease. This means that the gradient of the polynomial fits (lookup tables) can become extremely steep, resulting in essentially no variation in AoA for a varying mixing ratio (see Fig. 4). Lower mixing ratios, where the gradient becomes too steep, no longer carry information about AoA and are therefore no longer useful for its calculation. We consider the correlation of a trace gas with AoA to be at risk of breaking down if the gradient of the respective lookup table exceeds five times the average gradient in the AoA range between one and four years. If this condition is met in five out of nine consecutive mixing ratio bins, the correlation is considered to have broken down, and the series of bins in question, as well as all subsequent bins, are removed from the lookup table. This ensures that short intervals of steeper gradients are still included in the lookup tables*
> (Lines 224 to 232 in revised manuscript)

Given that all lookup tables in the study were updated, all figures in the original manuscript and supplement had to be updated for the revised manuscript and supplement as well. As all updated figures, including the one showing the AoA-profiles for the GLORIA-B flights (Fig. 10 in original manuscript), remain very similar to their original counterparts, no changes to the conclusions and text had to be made in regard to the redo of the lookup tables and figures. Despite the constriction of the lookup tables to regions of clear correlation between AoA amd mixing ratio, the AoA-profiles "GLORIA new method" and "CLaMS clock tracer" are still in good agreement above roughly 21 km for both flights, albeit within a higher uncertainty range, in the updated version of Fig. 10. The persistent good agreement between the two profiles led us to believe that the AoA-profiles "GLORIA new method" are correct after all, and do not just reflect the maximum AoA value of the CLaMS model. In order to back up our assessment, we looked at the AoA-profiles in the recently published study:

> *T. J. Schuck, J. Degen, T. Keber, K. Meixner, T. Wagenhäuser, M. Ghysels, G. Durry, N. Amarouche, A. Zanchetta, S. van Heuven, H. Chen, J. C. Laube, S. Baartman, C. van der Veen, M. E. Popa, and A. Engel. Measurement report: Greenhouse gas profiles and age of air from the 2021 hemera-twin balloon launch. EGUsphere, 2024:1–24, 2024. doi: doi:10.5194/egusphere-2024-3279. URL https://egusphere.copernicus.org/preprints/2024/egusphere-2024-3279/. in review*

As Schuck et al. [2024] also used balloon-based measurements from Kiruna, Sweden, in August 2021, they essentially sampled the same air masses that GLORIA-B observed during its first flight. The fact that Schuck et al. [2024] used different kinds of measurements (AirCore and cryogenic whole-air samples) and a different calculation method (the standard method described by Garny et al. [2024]), yet obtained very similar results for the same air masses, strongly indicates that our AoA profiles, "GLORIA new method", are plausible. The passage quoted above by the reviewer were rewritten as follows in the section 3.4 "Application to GLORIA balloon data" of the revised manuscript:

*The two profiles of "GLORIA-B new method" are also in good agreement with the recently published AoA profiles by Schuck et al. [2024]. In their study, Schuck et al. [2024] calculated AoA values using cryogenic whole-air samples of sulfur hexafluoride ($SF_6$) and carbon dioxide ($CO_2$) collected in Kiruna, Sweden, during August 2021. Similar to the AoA profiles of "GLORIA-B new method" the results reported by Schuck et al. [2024] exhibit an increase in AoA with altitude, reaching approximately five years at around $22\,km$. Beyond this altitude, the AoA values remain approximately constant at their maximum of five years. There are good reasons to believe that the utilized model-based lookup tables do reflect the actual atmospheric conditions well enough to justify their application to the GLORIA-B measurements. These reasons will be discussed in the next section. On top of that, an alternative to the model in the form of global satellite measurements as the foundation of the lookup tables will also be discussed in the next section. (Lines 475 to 483 in revised manuscript)*

The agreement between the AoA-profiles of "GLORIA-B new method" and the ones by Schuck et al. [2024] is further discussed in the newly added section 4.4 "Model and method uncertainties" in the revised manuscript:

*The excellent agreement between the AoA profiles reported by Schuck et al. [2024] and those derived using the proposed method from the GLORIA-B measurements at Kiruna (see Fig. 11). The balloon-borne air samples collected by Schuck et al. [2024] should closely resemble the air masses observed by GLORIA-B during its first flight, as both balloon flights took place in August 2021 in Kiruna, Sweden. The fact that Schuck et al. [2024] obtained similar results for comparable air masses using a different calculation method strongly supports our expectation that the utilized CLaMS-based lookup tables reflect the conditions of the actual atmosphere reasonably well.*

**MC2**

The Upper Boundary Condition (UBC) for SF6 plays an important role in this study where it is located at the stratopause (55km). This is explained (lines 145-147) as

"*The upper boundary condition for times outside of the measurement period was created by parameterizing the depicted seasonal cycle of each latitude with a sinusoidal least square fit and adding it to a shifted tropospheric tropical time series (taken from the lower boundary of SF6).*"

If that parameterization is as simple as described, I have some concerns. The Semi-Annual Oscillation already plays an important role at the stratopause, and it is modulated by the Quasi Biennal Oscillation (see e.g. Garcia et al., 1994). Or maybe that your sinusoidal least square fit includes several frequencies? Please clarify, and insert a reference if this fit was described in more detail in earlier work.

We added the following clarification for the way the upper boundary condition of $SF_6$ was created:

*The upper boundary condition for times outside of the measurement period was created by parameterizing the depicted seasonal cycle of each latitude with a sinusoidal least square fit of a single frequency and adding it to a shifted tropospheric tropical time series (taken from the lower boundary of $SF_6$)* (Lines 152 to 155 in revised manuscript)

We agree that the parameterisation of the upper boundary is rather simple, as it only includes an annual cycle and long-term trend outside the MIPAS measurement period. However, for our study this has no impact as we don't consider inter-annual variability here. It would be desirable to include more measurements beyond MIPAS and other variability modes in the parameterisation of the upper boundary in the future. A related clarification sentence has been added to the manuscript:

> *This simple parameterization could be further improved by including other modes of variability semi-annual oscillation [e.g. semi-annual oscillation, Garcia et al., 1997] which could be important for analysis of inter-annual variability, that is, however, not considered in this study.* (Lines 157 to 159 in revised manuscript)

Could you also be more specific with the UBC for HCFC-22 ? Line 155 states

> *"An open upper boundary condition has therefore been defined for HCFC-22."*

What does this mean? In my favorite (Eulerian) CTM, anything can happen at the upper boundary if no UBC is specified. Maybe a zero vertical gradient (i.e. null flux) is specified? These concerns could also be addressed by showing the timeseries of the (modelled and observed) $SF_6$ as well as the (modelled) HCFC-22 at the uppermost level.

The sentence above was specified as follows:

> *An open upper boundary condition (null flux) has therefore been defined for HCFC-22.* (Lines 166 to 167 in revised manuscript)

Concerns about the upper boundary conditions of the trace gases are understandable, as they can potentially lead to biases in the model results. In this specific case, however, the influence of the upper boundary is small, as air masses from the upper boundary region do not seem to get transported below 25 km in large enough quantities to meaningfully affect our analysis. This fact becomes evident by the newly added figure A4 (revised manuscript), which gets discussed in the newly added section 4.4 "Model and method uncertainties" (added under same name and number to this document).

**MC3**

What are the limitations of deriving LUT with only one model driven by one reanalysis dataset? Lines 443-444 state

> *"It should be noted that this slow bias of the ERA5 circulation in the comparison presented here is independent of the age calculation method used".*

...and I agree: a biased circulation in the driving reanalysis should impact the long-lived tracers and the clock tracer in the same manner, thus delivering the same correlations as a "perfect" reanalysis. On the other hand, the previous paragraph states (lines 435-436):

> *"...the correlations used in the proposed new age calculation method are based on model simulations and not necessarily reflect the actual atmospheric conditions."*

...and I also agree with this, but the two sentences seem to contradict each other. Hence I recommend to discuss these "existential questions" a little further in Section 4.

We appreciate the reviewer's insightful comments and agree that the apparent contradictions

highlighted in the original text need to be addressed. Additionally, we recognize the importance of discussing the limitations of the derived lookup tables (LUTs). Upon further consideration, and in light of the strong agreement between the AoA profiles derived using the improved lookup tables ("GLORIA new method") and the independent AoA profiles reported by Schuck et al. [2024], we have revised our perspective. This agreement has increased our confidence that the LUTs reflect actual atmospheric conditions reasonably well. As a result, we have removed the statement that the utilized lookup tables "might not perfectly reflect the actual atmospheric conditions" from the revised manuscript.

In the new discussion section, "4.4 Accuracy of the CLaMS Model" (added to this document under the same number and name), we outline three reasons for our confidence in applying the CLaMS-based lookup tables to the GLORIA measurement data. Nevertheless, we emphasize that this study is primarily a proof of concept and that a thorough analysis to quantify the influence of model parameters on the AoA analysis is necessary. However, such an analysis exceeds the scope of this study.

**Specific Comments**

**SC1**

Terminology: some recurring words lead to confusion and should be replaced throughout the text:

- If I understand well, the midpoints of the AoA/vmr bins are interpolated with a polynomial fit which uses the Savitzky-Golay-Filter to create smoother LUT. The result is variously described as *"interpolated series"*, *"interpolation"*, *"smoothed series"*, *"polynomial fit"* or *"Savitzky-Golay-Filter"*. Please use a consistent term throughout the text and in the figures (*"polynomial fit"* is fine) and provide a reference on the Savitzky-Golay-Filter or its usage for this type of procedure (line 206).

  The terms *"interpolated series"* and *"smoothed series"* have been replaced with *"polynomial fit"*, which is now used more consistently throughout the revised manuscript, particularly in section 2.2, *Age of Air Calculation.* The following sentence has been added to reference the applied Savitzky-Golay filter:

  > *The Savitzky-Golay filter (`scipy.signal.savgol_filter`) was implemented using the SciPy library [Virtanen et al., 2020].*
  > (Lines 220 to 221 in revised manuscript)

- The CLaMS AoA derived from the lag-time of the clock tracer is written very often as the *"true" AoA* or the *"exact" AoA*. Even with the quotes, this is misleading since this AoA is biased by ERA-5... In most cases no adjectives are necessary as it is perfectly clear which AoA you are using as reference. But if an adjective is necessary, I recommend to write systematically *"the **actual** AoA from the clock-tracer"*.
  The terms *"true AoA"* and *"exact AoA"* have been consistently replaced with the suggested term *"actual AoA from the clock-tracer"* throughout the revised manuscript.

- From lines 211 to 308 there are 5 attributions of AoA spread in a mixing ratio bin to *"natural variability"*. This is misleading for the same reason: this variability arises from the model simulation and its driving reanalysis and is necessarily natural... Hence I recommend to replace *"natural variability"* by *"**model** variability"* or *"**model/reanalysis** variability"*.
  The term *"natural variability"* has been consistently replaced with the suggested term *"model variability"* throughout the revised manuscript.

- *"one sigma environment(s)"* → *"**one-sigma range(s)**"*
  The term *"one sigma environment(s)"* has been consistently replaced with the suggested term *"one-sigma range(s)"* throughout the revised manuscript.

**SC2**

Abstract, line 6:
*"this method works well **in most of the lower stratosphere** up to a height of about 25 km"* (in order to account for the white areas in Fig. 6 to 9).
The sentence was changed accordingly in the abstract of the revised manuscript.

**SC3**

Abstract, line 8-9:
*"The **multi-species** age calculation method is evaluated in a model environment and compared against the  **actual** model age from an **idealized clock tracer**."*
The sentence was changed accordingly in the abstract of the revised manuscript.

**SC4**

The introduction is well written, but please cite some previous papers using compact correlations between AoA and long-lived tracers to derive an estimation of the AoA, and explain how their aims differ from those in your study. To the best of my knowledge, the first such paper was written by **Linz et al. (2017)** and the latest one was written by **Dubé et al. (2024)**. Even though the latter is still a preprint, it is important to cite it because it also uses CLaMS.

Part of the introduction was rewritten as follows to include the two mentioned studies:

> *This study presents a new method to derive AoA from atmospheric measurements of multiple long-lived trace gases. The method is based on the correlations of the trace gas species $CCl_3F$ (trichlorofluoromethane (CFC-11)), $CCl_2F_2$ (CFC-12), $CHClF_2$ (HCFC-22), methane ($CH_4$), nitrous oxide ($N_2O$), and $SF_6$ with AoA. The application and evaluation of the method are carried out using simulations with the chemistry transport model CLaMS (Chemical Lagrangian Model of the Stratosphere). Notably, a recent study by Dube et al. [2024] also utilizes the tight correlations between long-lived trace gases and AoA in the CLaMS model. However, their focus is on deriving decadal trends in AoA from these correlations rather than AoA itself. Additionally, Linz et al. [2017] used $N_2O$ in addition to $SF_6$ to calculate AoA. They derived AoA from $N_2O$ using an empirical relationship between AoA and $N_2O$ that was calculated from balloon and aircraft measurements. Their study leveraged AoA to calculate the strength of the total overturning circulation through different isentropes, offering insights into large-scale atmospheric dynamics.*
>
> *In this study, however, a weighted mean AoA is calculated from modeled satellite measurements of the aforementioned six trace gases. The difference between this weighted mean AoA and the actual AoA from the clock-tracer in the model is found to remain below half a year in the lower stratosphere. In addition, the proposed method is also applied to measurements with the GLORIA-B instrument, yielding more plausible results and drastically reduced uncertainty compared to conventional methods.*
>
> (Lines 82 to 95 in revised manuscript)

**SC5**

Section 2.1: please provide some details about modelling of vertical circulation in CLaMS. I guess that these are radiative heating rates between the isentropes, but do they come from ERA-5 or from an internal part of CLaMS? This is important as in each case there could be biases in these heating rates (which could be mentioned later in the discussion of the results).

The corresponding passage in the original manuscript (Sect 2.1, Line 179) was rewritten as follows for clarification:

> *The vertical cross-isentropic model transport in CLaMS was calculated from the European Centre for Medium-Range Weather Forecasts Reanalysis v5 (ERA5) total diabatic heating rates (see Ploeger et al. [2021]).*
>
> (Lines 173 to 174 in revised manuscript)

**SC6**

Legend of Figure 2: the details of this Figure are already explained in its lower text box and in the text, so I believe that it is not necessary to repeat them for a third time in this Legend (also because all terminology issues arise there - see SC1). A shorter legend could be:

*"**Figure 2.** Schematic representation of method used to estimate the **AoA corresponding to measured mixing ratios and its associated** uncertainty **(see text for details)**."*

The legend of the figure was changed accordingly (Fig. 2 in revised manuscript).

**SC7**

Line 219: *". . . added together through means of (Gaussian) error propagation. . ."*
→ do you mean that they are added as a Root Mean Square Error (RMSE) ?

The term *"Gaussian) error propagation"* was indeed used to refer to the Root Mean Square Error (RMSE) of two independent uncertainty values. The sentence was rephrased as follows for clarification:

> *The model variability and the converted measurement uncertainty added together as a Root Mean Square Error (RMSE) yield the total uncertainty of the AoA obtained for the measured mixing ratio.*
> (Lines 243 to 245 in revised manuscript)

**SC8**

Line 268, first sentence introducing Fig 3: please clarify here that you are not yet showing the binning hemispheric approach described in the earlier section 2.2.

The following sentence has been added in the revised manuscript:

> *Note that all panels of the figure show the results for both hemispheres together.*
> (Lines 293 to 294 in revised manuscript)

**SC9**

Line 270:
*"Additionally, the equivalent latitudes of the model points **(Nash et al., 1996)** are color-coded. . ."*

The missing reference (Nash et al., 1996) was added as suggested (line 296 in the revised manuscript)

**SC10**

Lines 312-317: this procedure is written in a rather obscure manner, even though it seems quite simple. Please re-formulate, e.g. explaining that prior to the application of these LUT you want to compare the 6 total uncertainties for each species among each other and also with the total combined uncertainty.

The procedure was not described in an easy-to-understand manner. It was rephrased as follows:

> *In the case of actual measurements, AoA and its total uncertainty would be interpolated from the lookup tables for each of the six trace gases separately and the weighted mean $\overline{AoA}$ (see Eq. 1) with the combined uncertainty $\overline{\sigma}$ (see Eq. 2) would subsequently be constructed from the results. Before calculating the actual mean AoA values, an initial estimate of the combined uncertainty for AoA values ranging from*

*one to five and a half years was made. This involved assuming that a specific AoA value within the range was uniformly derived from the lookup tables of all six trace gases, implying that all tracers yielded the same AoA value. This value, treated as the weighted mean AoA, allowed for interpolation of corresponding total uncertainties from each tracer's lookup table. These uncertainties were then used to construct a combined uncertainty $\bar{\sigma}$.*
(Lines 344 to 350 in revised manuscript)

**SC11**

Line 337: "*(compare **with** Fig. A4)*". This figure A4 is compared multiple times with Fig. 5, 6, 7. I think that it should be moved from the Appendix to the main text.

The figure in question was moved from the Appendix to the main article (Fig. 6 in revised manuscript).

**SC12**

Lines 355-356: "*The pseudo-measurements were created by adding normally distributed random noise to the mixing ratios of the six trace gases for all air parcels on the four considered days.*"

These synthetic measurements are not completely realistic as they do not take systematic errors into account. This is a real concern and should be mentioned somehow, because actual measurements would be biased w.r.t. modeled mixing rations since there are biases in the ERA-5 circulation.

It was not specifically stated that the estimated uncertainties take systematic errors into account. The paragraph first mentioning the uncertainties in Sect. 2.2 "*Age of air calculation*" was rewritten as follows:

*For the measurement uncertainties of the six trace gases, the values listed in Tab. 2 are used. These values represent estimations of the total uncertainties for the measurement of the six trace gases with the proposed Changing-Atmosphere Infra-Red Tomography Explorer (CAIRT) instrument. These values are representative for a typical satellite instrument and take into account both the detector noise as well as biases e.g. from in-flight calibrations [ESA, 2023]. It is expected that this is a reasonable reflection of the true uncertainty as long as the AoA inferred from several trace species are combined and the uncertainty would be combined, but likely is an underestimate where only one or two trace-species remain to have meaningful correlations.*
(Lines 236 to 241 in revised manuscript)

**SC13**

SC13. Line 362: "*The thick black lines represent the zonally averaged tropopause from the ERA5 reanalysis data for each day*". Is this the thermal (T gradient) or dynamical (PV+theta) tropopause?

For clarification, the sentence above was rewritten as follows in the revised manuscript:

*The thick black lines represent the zonally averaged tropopause, derived from the daily ERA5 reanalysis data (lapse rate tropopause following WMO (1957)).* (Lines 394 to 396 in revised manuscript)

**SC14**

Lines 365-367:

> *"Even the northward intrusion of air with AoA below one and a half years into the layer of air with AoA between one and a half and two years at roughly 70°N and 14 km height in July is clearly present in the results of the correlation method (compare Figs. A4 (c) and 7 (c))"*

This region is not obvious to identify at first. Consider helping the reader by drawing a bounding box around it, at least for these two figures.

A corresponding bounding box was added to the two figures (Figs 6 and 8 in revised manuscript) to highlight the region in question.

**SC15**

Lines 385-386: you mention Figure 8 but shouldn't that be Fig. 9 there ?
Figure 8 was referenced correctly, but the corresponding paragraph could have been written more clearly. Here is the rewritten version of the revised manuscript:

> *Figure 10 presents the standard deviations in the zonal direction of the differences between pseudo-measurement and clock tracer AoA. These standard deviations quantify the spread of AoA differences in the zonal direction and can be used to estimate the uncertainty of AoA derived from individual measurements at different longitudes for any latitude. If the zonally averaged differences in AoA shown in Fig. 9 were uniformly zero, the standard deviations in Fig. 10 could directly represent these uncertainties. In such a scenario, Fig. 10 would closely resemble the initial uncertainty estimate in Fig. 7. However, as the differences in Fig. 9 are not uniformly zero, the two figures differ slightly. Specifically, the values in Fig. 9 are higher near the tropopause and towards the top of the height scale across all four days. To estimate the total uncertainty of AoA derived from individual measurements, the zonally averaged differences in AoA and the standard deviations in Fig. 10 must be combined as a Root Mean Square Error. This calculation yields an average uncertainty of approximately 0.3 years for AoA at any latitude and longitude across the four days. A graphical depiction of the expected uncertainty for AoA derived from individual measurements is provided in Fig. S29 in the supplementary material.* (Lines 412 to 422 in revised manuscript)

**SC16**

Lines 385-388:

> *"(Note that the scales of the colorbars are different and that the contours show different things in the two figures)"*

This is the straw that broke the camel's back. I was wondering since Fig. 8 about the added value in showing absolute differences with color coding and relative differences with contour labels, which is quite confusing and was not done for fig. 6. Note that fig. 6 also could have shown absolute errors in color shading and relative errors with contour labels, but I am not complaining that this was not done. On the contrary, I really think that the color scales and/or contour labels should be changed in Fig. 8 and 9 to simplify them and allow direct comparison with fig. 6 !

We wanted to add additional information to the figures by displaying relative values with contours and absolute values with color-coding. We understand that we have created unnecessary confusion, and changed the contours in all figures to display the absolute values of the color-coding instead. We also realize that different color-scales aren't ideal for comparing figures and adjusted the color-scale of Fig. 9 to match that of Fig. 6 (Figs. 7 and 10 in revised manuscript)

**SC17**

Lines 395-401: please synthesize in a few sentences

> "...the standard convolution method, as described in Garny et al. [under review], and the subsequent correction for SF6-depleted air from the mesosphere introduced by Garny et al. (2024)."

... including the limitations of this method and this correction. This is important because these limitations play a role in the discussion of Fig. 10 while the review by Garny et al. is still being reviewed.

The following passage has been added to the section 3.4 "Application to GLORIA balloon data":

> In the standard convolution method, the mean age of air spectrum is modeled using an inverse Gaussian distribution. This distribution is fully defined by its first two moments: the mean (AoA) and the width ($\Delta$). Alternatively, the inverse Gaussian can be specified using AoA and the ratio of its first two moments ($rom = \frac{AoA}{\Delta}$). Importantly, the ratio of moments remains relatively stable over time at a given location and can be determined from model-based or measurement-based lookup tables. Once the ratio of moments is known, the inverse Gaussian can be constructed for a range of AoA values. The method involves convolving the tropospheric $SF_6$ time series with the generated inverse Gaussians. The AoA value corresponding to the convolution result that best matches an observed $SF_6$ mixing ratio is then selected as the final estimate. Due to the unimodal nature of the inverse Gaussian distribution, the standard convolution method cannot resolve additional modes in the mean age of air spectrum that may arise from the seasonality of the Brewer–Dobson circulation (BDC). The AoA correction for $SF_6$-depleted air from the mesosphere is derived using a least-squares fit of an exponential function to global model results, establishing a relationship between uncorrected and actual AoA. While the uncertainty of this correction method increases with AoA, it is optimized for AoA values up to 5 years. Despite this limitation, the method continues to enhance the accuracy of results even for AoA values exceeding 5 years.   (Lines 428 to 439 in revised manuscript)

**SC18**

Figure 10: the dashed "*greenish*" lines and the greenish shadings are not really visible, especially on top of the blue shadings. Consider using black lines and black horizontal error bars instead (possibly for a subset of the levels in the case of horizontal error bars).

The color of the profile and its uncertainty range were recolored black for better visibility in the revised manuscript (Fig. 11).)

**SC19**

Lines 430-432: " *Another possiblye explanation  **would** be  some issue with the instrument during the Timmins flight leading to the retrieval of systematically too low SF6 mixing*

ratios. *Perhaps the unintended descend t of the gondola down to 22 km mentioned in sect. 2.3 could have caused such an issue.*"

→ . . . while not damaging the retrievals of the 5 other species, leading to the very different values by "GLORIA new method" ??

The passage above has been rephrased as follows:

> *Another possible explanation would be some issue with the instrument during the Timmins flight that led to the retrieval of systematically too low $SF_6$ mixing ratios. There is, however, no indication of what could have caused such an issue thus far.* (Lines 471 to 473 in revised manuscript)

**SC20**

Line 462: "*The lookup tables for the remaining cases can be found in Figs. S17 to S28 of the supplement to this article*" → these are not LUT, they are plots of the LUT (see MC1).

The sentence has been rewritten for clarification in the revised manuscript as follows:

> *Graphical depictions of the lookup tables for the remaining cases can be found in Figs. S17 to S28 of the supplement to this article.* (Lines 507 to 508 in revised manuscript)

**SC21**

Figure 11(f): "*...by means of gaussian error propagation). Contours: Relative difference between color-coded values and AoA from clock tracer.*"
This is very unclear... What do these contour lines mean exactly? Are they necessary? They are not discussed in the text...

In response to **SC7** and **SC16** the phrase *gaussian error propagation* was replaced by Root Mean Square Error, and the contours in all figures changed to reflect the same values as the corresponding color-coding. The caption of the corresponding figure (Fig. 12 in revised manuscript)

> *Selection of plots exemplifying the analysis in Sec. 3.3 for lookup tables created from zonally averaged Chemical Lagrangian Model of the Stratosphere (CLaMS) results. All plots refer to 1 July 2011. Bold black line in (c) to (f) indicates zonally averaged tropopause according to ERA5 reanalysis. **(a)** Creation of lookup table for zonal mean AoA from zonal mean CFC-12 in southern hemisphere (see Figs. 4 and A2). **(b)** Dashed/dotted/dash-dotted colored lines: Total uncertainty of each tracer in southern hemisphere. Solid black line: Combined total uncertainty for all tracers according to Eq. 2 (see Figs. 5 and A3). **(c)** Zonally averaged weighted mean AoA from synthetic measurements with lookup tables (see Fig. 8). **(d)** Zonally averaged difference between weighted mean AoA from synthetic measurements and CLaMS clock tracer (see Fig. 9). **(e)** Standard deviations of zonally averaged differences between weighted mean and clock tracer AoA (see Fig. 10). Note that the differences are calculated before the standard deviations. **(f)** Estimation of uncertainty of weighted mean AoA from lookup tables along all longitudes (values in (d) and (e) added together as a RMSE).*

**SC22**

Section 4.2: this discussion on the stability of the correlations is quite confusing:
"*Since these depletion processes do not fundamentally change over time, the future correlations of the five mentioned trace gases with AoA will likely be similar to the way they were in 2011, the year considered in this study*".
But GLORIA-B used LUT made specifically for its flight dates, so this question about the stability of the correlations is moot: the model correlations can be computed for the days of the measurements, i.e. in much more recent years tha 2011, and this was actually done! It looks to me like this section was written before the comparison with GLORIA-B and not removed afterwards.

Section 4.2 "Stability of correlations" has been removed from the revised manuscript

**SC23**

Line 574: " *Also, such datasets could be used to study exchange processes between the troposphere and the stratosphere and therefore help to better constrain new emissions of prohibited substances like CFC's*".
This sentence is not clear. There are many other good reasons to sudy troposphere-stratosphere exchanges.

We agree that there are indeed many important reasons to study these exchanges beyond constraining emissions of prohibited substances. In light of your feedback, we have revised the sentence to provide a more comprehensive view of the applications of such studies. The revised version now reads:

> *Furthermore, such datasets could be invaluable for studying exchange processes between the troposphere and the stratosphere. These studies have wide-ranging applications in atmospheric science, including but not limited to: understanding the transport of water vapor and other trace gases, investigating the impact of stratospheric ozone on climate, assessing the influence of tropospheric pollutants on stratospheric chemistry, and helping to better constrain emissions of various substances, including prohibited ones like CFCs.* (Lines 643 to 647 in revised manuscript)

**Typos, wording etc.**

- Line 18: "... Brewer-Dobson circulation (**BDC; see e.g.** Holton et al., 1995; Butchart, 2014, )

- Line 78: "... if the**y** can be retrieved. . ."

- Lines 198-205: " **For visual clarity**, *only the last three of these mixing ratio bins are shown in the figure. The histogram illustrates the distribution of AoA within a given mixing ratio bin. The blue area in the histogram highlights the one sigma environment around the mean AoA value of the distribution (mean value ± one*  **standard deviation**). *Similarly, the blue area in each of the three depicted mixing ratio bins corresponds to the one-sigma*  **range** *around the mean value of the respective AoA distribution inside. Such one-sigma*  **ranges** *are calculated for each one of the one hundred fifty mixing ratio bins. Subsequently, a midpoint for each bin with the sample mean AoA as the x- and the middle of the bin range as the y-coordinate was then defined. Th**is**e  constructed set of midpoints constitutes a sort of look-up table that can be used to interpolate a**n** AoA value. . .*"

- Line 216: first sentence actually belongs to previous paragraph.

- Lines 372: "*The absolute difference  reach**es** its highest values...*"

- Lines 381-382: "*These standard deviations are a quantification of the spread of the AoA difference in zonal direction. They can be used to estimate the uncertainty of AoA derived from individual measurements at different  **longitudes** for any  **latitude**.*"

- Line 401: first sentence actually belongs to previous paragraph.

- Line 410: "The magenta shading around the "GLORIA-B new method " AoA represents..."

- Line 425: "...however, any conclusions drawn from the  **corresponding** values are hardly meaningful"

- Lines 451-472: please replace all occurrences of "3d" with "3-D".

We have thoroughly corrected all the spelling mistakes highlighted and implemented all the suggested word changes throughout the revised manuscript.

**Additional Bibliographical References**

- Garcia, R. R., et al. (1997). *Climatology of the semiannual oscillation of the tropical middle atmosphere. J. Geophys. Res.* DOI: 10.1029/97JD00207.

- Nash, E. R., P. A. Newman, J. E. Rosenfield, and M. R. Schoeberl (1996), An objective determination of the polar vortex using Ertel's potential vorticity, J. Geophys. Res., 101(D5), 9471–9478, doi:10.1029/96JD00066.

- Linz, M., et al. (2017). *The strength of the meridional overturning circulation of the stratosphere. Nature Geoscience.* DOI: 10.1038/ngeo3013.

- Dubé, K., et al. (2024). *Hemispheric asymmetry in recent stratospheric age of air changes. EGUsphere [preprint].* DOI: 10.5194/egusphere-2024-1736.

We have included the additional references mentioned above into the revised manuscript

**4 Discussion**

**4.4 Model and method uncertainties**

This study should, first and foremost, be viewed as a proof of concept for the proposed method to calculate AoA. Such a proof of concept can only be demonstrated within the perfectly self-consistent environment of a model, where the "true" values of the desired quantity are already known. Since no model is a perfect representation of the actual atmosphere, the lookup tables required for applying the proposed method should ideally be constructed directly from measurements, as laid out in Sect. 4.1, rather than from model results. Due to the lack of suitable measurement data, we had to establish the lookup tables required for applying the method to the GLORIA-B data using the available model results. We are confident that the CLaMS-based lookup tables used on the GLORIA-B data represent the actual atmospheric conditions reasonably well and find the AoA profiles derived from these lookup tables to be more plausible than the corresponding ones derived with the standard method (see Fig. 11). Our confidence in the CLaMS-based lookup tables stems from the following three reasons:

1. The reliability of the CLaMS model for the utilized trace gases has been demonstrated in previous studies (see, e.g., Laube et al. [2020] for CFC-11, CFC-12, and HCFC-22, Konopka et al. [2004] for $CH_4$, and Pommrich et al. [2014] for $N_2O$).

2. There is excellent agreement between the AoA profiles reported by Schuck et al. [2024] and those derived using the proposed method from the GLORIA-B measurements at Kiruna (see Fig. 11). The balloon-borne air samples collected by Schuck et al. [2024] should closely resemble the air masses observed by GLORIA-B during its first flight, as both balloon flights took place in August 2021 in Kiruna, Sweden. The fact that Schuck et al. [2024] obtained similar results for comparable air masses using a different calculation method strongly supports our expectation that the utilized CLaMS-based lookup tables reflect the conditions of the actual atmosphere reasonably well.

3. The upper boundary region, arguably the biggest source of bias in the simulation as a whole, seems to have only limited impact on the model tracer mixing ratios in the considered altitude region (i.e., below $25\,\mathrm{km}$). Figure A4 in the appendix shows the temporally averaged zonal mean $SF_6$ distribution of the model results up to the upper boundary over two different three-month periods. Fig. 11a shows the average distribution from December 2010 to February 2011, representing winter in the Northern Hemisphere, and Fig. 11b shows the average distribution from June 2011 to August 2011, representing winter in the Southern Hemisphere. The winter periods were chosen because they represent the time when the downward transport of air from higher altitudes by the deep branch of the BDC is strongest in the respective hemisphere. $SF_6$ was selected because, in relative terms, it reaches the upper boundary in higher amounts than all other tracers (due to the absence of a stratospheric sink). Out of all model results, the influence of the upper boundary is therefore expected to be strongest for $SF_6$ during winter in the respective hemisphere. Figure 11 suggests that the downward transport 11, the downward transport of $SF_6$ depleted air from the upper boundary only seems to have an affect below $25\,\mathrm{km}$ at high latitudes in the winter hemisphere. This limited influence of the upper boundary on model results below $25\,\mathrm{km}$ could partly be related to the slow bias of the stratospheric ERA5 circulation, as pointed out in sect. 3.4. This slow bias, however, effects the six long-lived trace gases in the same way as the model clock tracer and therefore can't be responsible for any biases the model-based lookup tables might have.

While these considerations give us confidence in the plausibility of the CLaMS-based lookup tables, it is important to emphasize that a thorough analysis would be necessary to fully assess

the influence of the upper boundary on each of the six trace gases specifically, and to evaluate the extent to which the derived lookup tables reflect the conditions of the real atmosphere. Such an analysis would ideally involve the use of several atmospheric models and/or reanalysis datasets to better quantify potential biases and uncertainties. Additionally, the influence of uncertainties in the kinetic model parameters, such as photochemical data and model-calculated loss rates, on AoA cannot be ruled out. A thorough analysis of the influence of these uncertainties on AoA would also be needed. However, conducting such extensive analyses is beyond the scope of this study.

**References**

K. Dube, S. Tegtmeier, F. Ploeger, and K. A. Walker. Hemispheric asymmetry in recent stratospheric age of air changes. EGUsphere, 2024:1–21, 2024. doi:10.5194/egusphere-2024-1736. URL `https://egusphere.copernicus.org/preprints/2024/egusphere-2024-1736/`.

H. Garny, F. Ploeger, M. Abalos, H. Bönisch, A. E. Castillo, T. von Clarmann, M. Diallo, A. Engel, J. C. Laube, M. Linz, J. L. Neu, A. Podglajen, E. Ray, L. Rivoire, L. N. Saunders, G. Stiller, F. Voet, T. Wagenhäuser, and K. A. Walker. Age of stratospheric air: Progress on processes, observations, and long-term trends. Reviews of Geophysics, 62(4):e2023RG000832, 2024. doi:https://doi.org/10.1029/2023RG000832. URL `https://agupubs.onlinelibrary.wiley.com/doi/abs/10.1029/2023RG000832`. e2023RG000832 2023RG000832.

Marianna Linz, R Alan Plumb, Edwin P Gerber, Florian J Haenel, Gabriele Stiller, Douglas E Kinnison, Alison Ming, and Jessica L Neu. The strength of the meridional overturning circulation of the stratosphere. Nature Geoscience, 10(9):663–667, 2017.

F. Ploeger, M. Diallo, E. Charlesworth, P. Konopka, B. Legras, J. C. Laube, J.-U. Grooß, G. Günther, A. Engel, and M. Riese. The stratospheric Brewer–Dobson circulation inferred from age of air in the ERA5 reanalysis. Atmos. Chem. Phys., 21(11):8393–8412, 2021. doi:10.5194/acp-21-8393-2021. URL `https://acp.copernicus.org/articles/21/8393/2021/`.

T. J. Schuck, J. Degen, T. Keber, K. Meixner, T. Wagenhäuser, M. Ghysels, G. Durry, N. Amarouche, A. Zanchetta, S. van Heuven, H. Chen, J. C. Laube, S. Baartman, C. van der Veen, M. E. Popa, and A. Engel. Measurement report: Greenhouse gas profiles and age of air from the 2021 hemera-twin balloon launch. EGUsphere, 2024:1–24, 2024. doi:10.5194/egusphere-2024-3279. URL `https://egusphere.copernicus.org/preprints/2024/egusphere-2024-3279/`. in review.